# An inflammatory-CCRK circuitry drives mTORC1-dependent metabolic and immunosuppressive reprogramming in obesity-associated hepatocellular carcinoma

Hanyong Sun[1,2], Weiqin Yang[3], Yuan Tian[1,4], Xuezhen Zeng[3], Jingying Zhou[3], Myth T.S. Mok[3], Wenshu Tang[3], Yu Feng[3], Liangliang Xu[3], Anthony W.H. Chan[5], Joanna H. Tong [5], Yue-Sun Cheung[6], Paul B.S. Lai[6], Hector K.S. Wang[7], Shun-Wa Tsang[7], King-Lau Chow[7], Mengying Hu[8], Rihe Liu[9], Leaf Huang[8], Bing Yang[4], Pengyuan Yang[4], Ka-Fai To[5,10], Joseph J.Y. Sung[1,11], Grace L.H. Wong[1,11], Vincent W.S. Wong [1,11] & Alfred S.L. Cheng[3,11]

Obesity increases the risk of hepatocellular carcinoma (HCC) especially in men, but the molecular mechanism remains obscure. Here, we show that an androgen receptor (AR)-driven oncogene, cell cycle-related kinase (CCRK), collaborates with obesity-induced pro-inflammatory signaling to promote non-alcoholic steatohepatitis (NASH)-related hepatocarcinogenesis. Lentivirus-mediated *Ccrk* ablation in liver of male mice fed with high-fat high-carbohydrate diet abrogates not only obesity-associated lipid accumulation, glucose intolerance and insulin resistance, but also HCC development. Mechanistically, CCRK fuels a feedforward loop by inducing STAT3-AR promoter co-occupancy and transcriptional up-regulation, which in turn activates mTORC1/4E-BP1/S6K/SREBP1 cascades via GSK3β phosphorylation. Moreover, hepatic *CCRK* induction in transgenic mice stimulates mTORC1-dependent $G-csf$ expression to enhance polymorphonuclear myeloid-derived suppressor cell recruitment and tumorigenicity. Finally, the STAT3-AR-CCRK-mTORC1 pathway components are concordantly over-expressed in human NASH-associated HCCs. These findings unveil the dual roles of an inflammatory-CCRK circuitry in driving metabolic and immunosuppressive reprogramming through mTORC1 activation, thereby establishing a pro-tumorigenic microenvironment for HCC development.

[1] Department of Medicine and Therapeutics, The Chinese University of Hong Kong, Hong Kong, China. [2] Department of Liver Surgery, Ren Ji Hospital, School of Medicine, Shanghai Jiao Tong University, Shanghai 200127, China. [3] School of Biomedical Sciences, The Chinese University of Hong Kong, Hong Kong, China. [4] Key Laboratory of Infection and Immunity, Institute of Biophysics, Chinese Academy of Sciences, Beijing 100101, China. [5] Department of Anatomical and Cellular Pathology, The Chinese University of Hong Kong, Hong Kong, China. [6] Department of Surgery, The Chinese University of Hong Kong, Hong Kong, China. [7] Division of Life Science, The Hong Kong University of Science and Technology, Hong Kong, China. [8] Division of Pharmacoengineering and Molecular Pharmaceutics, Eshelman School of Pharmacy, University of North Carolina at Chapel Hill, Chapel Hill, NC 27599, USA. [9] Division of Chemical Biology and Medicinal Chemistry, Eshelman School of Pharmacy and Carolina Center for Genome Sciences, University of North Carolina at Chapel Hill, Chapel Hill, NC 27599, USA. [10] State Key Laboratory of Translational Oncology, The Chinese University of Hong Kong, Hong Kong, China. [11] State Key Laboratory of Digestive Disease, The Chinese University of Hong Kong, Hong Kong, China. These authors contributed equally: Hanyong Sun, Weiqin Yang, Yuan Tian, Xuezhen Zeng. Correspondence and requests for materials should be addressed to G.L.H.W. (email: wonglaihung@cuhk.edu.hk) or to V.W.S.W. (email: wongv@cuhk.edu.hk) or to A.S.L.C. (email: alfredcheng@cuhk.edu.hk)

Hepatocellular carcinoma (HCC) is among the most lethal cancers that significantly correlate with obesity[1–3]. The pathophysiology begins with obesity-induced hepatosteatosis and non-alcoholic steatohepatitis (NASH), collectively known as non-alcoholic fatty liver disease (NAFLD), which can further develop into cirrhosis and HCC[4]. Notably, HCC is characterized by strong sexual dimorphism in almost all geographic areas where male to female ratios average between 2:1 and 7:1[5,6]. In a prospective study of >900,000 US adults, men with a body mass index (BMI) of 35 kg/m² exhibited a dramatic 4.52-fold increase in relative risk of death from liver cancer, while a modest 1.68-fold increase was observed in women[2]. A recent population-based cohort study of 5.24 million adults in United Kingdom confirmed the significant modulation of HCC incidence by gender, in which higher BMI in men but not in women was associated with substantially increased risk of HCC[1]. In addition, another population-based cohort study of 1.2 million Swedish men further showed that a high BMI (≥30 kg/m²) in late adolescence was associated with an increased risk of future severe liver diseases including HCC[3]. These findings consistently underscore the sex disparity in obesity-associated HCC, but the molecular mechanisms underlying HCC development in obese men remain obscure[4,6].

Using obese mouse models exposed to the hepatic procarcinogen diethylnitrosamine (DEN), Park et al. demonstrated that obesity is a bona fide liver tumor promoter[7]. The obesity-driven HCC development largely depends on a chronic pro-inflammatory state that results in elevated circulating levels of cytokines, such as tumor necrotic factor-α (TNF-α) and interleukin-6 (IL-6)[7,8], and the latter of which has recently been shown to correlate with HCC progression in obese people[9]. Chronic IL-6-mediated activation of signal transducer and activator of transcription 3 (STAT3) can cause hepatic insulin resistance critical for the development of glucose intolerance and steatotic HCC[10,11]. Unlike early hepatocarcinogenesis which relies on paracrine nuclear factor kappa B (NF-κB)-regulated IL-6 production by inflammatory cells[12], HCC progenitor cells in premalignant lesions acquire autocrine IL-6-STAT3 signaling to stimulate cellular proliferation and transformation[13]. Nevertheless, it is unclear how the hepatic IL-6-STAT3 cascade is activated and sustained during malignant transformation.

One of the major IL-6-driven signaling pathways in obesity and HCC development is mechanistic target of rapamycin (mTOR)[7], which is a key signal transducer in the phosphatidylinositol-4,5-bisphosphate 3-kinase (PI3K)/Protein Kinase B (AKT) pathway. mTOR can assemble with Raptor and Rictor to form two functionally distinct complexes, mTORC1 and mTORC2, respectively. Activation of cap-dependent translation by phosphorylation of 4E-BP1 contributes to mTORC1-dependent carcinogenesis[14]. Consistent with the increased de novo lipid synthesis in proliferating cancer cells, mTORC1 has been shown to activate the central lipogenic transcription factor, sterol regulatory element-binding protein 1 (SREBP1), through S6K1 to stimulate lipogenesis and cell proliferation[15]. Animal model and human studies have confirmed the functional significance of mTORC1 activation in NAFLD pathogenesis[7,16]. Stimulation of AKT-mTORC1 signaling, either alone[17] or in combination with β-catenin[18], induces hepatic lipogenesis and tumorigenesis. Nonetheless, how mTORC1 remains constitutively active in the context of insulin resistance is unresolved[19]. Additionally, mTORC1 was shown to be negatively regulated by glycogen synthase kinase 3β (GSK3β) via phosphorylation of tuberous sclerosis complex 2 (TSC2)[20], which transmits diverse upstream signals including insulin to mTORC1[21]. Moreover, inactivation of GSK3β was shown to inhibit hepatocellular apoptosis in dietary obesity-promoted HCC[22]. While these findings implicate a causal effect of GSK3β dysregulation in obesity-related hepatocarcinogenesis, the upstream kinase that controls GSK3β/mTORC1 signaling in the obesity-induced inflammatory microenvironment has not been elucidated.

Genetic and biochemical studies have demonstrated the fundamental roles of androgen receptor (AR) in male predominance of HCC[23]. Using genome-wide location and functional analysis, cell cycle-related kinase (CCRK), the latest cyclin-dependent kinase member (CDK20), was previously underpinned as a direct AR-regulated oncogene in hepatocarcinogenesis through concordant activation of GSK3β/β-catenin and AKT/EZH2 signaling[24,25]. It was further shown that CCRK mediates virus–host signaling to promote hepatitis B virus (HBV)-associated hepatocarcinogenesis[26], and fosters an immunosuppressive microenvironment for HCC development by induction of myeloid-derived suppressor cells (MDSCs)[27]. Notably, over-expression and hyper-activation of CCRK distinguishes a subset of HCC patients with poor overall and disease-free survival[24–27]. Based on the biological and clinical significance of CCRK in the male-predominant HBV-associated HCC[28], here we investigated its potential role in obesity-associated HCC. Our findings demonstrated that knockdown of Ccrk dramatically suppressed hepatic lipid accumulation, inflammation and tumorigenicity in multiple murine NASH and HCC models. Mechanistically, obesity-induced pro-inflammatory STAT3 and AR-induced signaling cooperatively up-regulated CCRK expression, which in turn activated the mTORC1 pathway crucial for lipid/glucose homeostasis, immunosuppression, and tumorigenesis. These findings suggest that CCRK functions as a major signaling hub in obesity-associated hepatocarcinogenesis, providing insights into therapeutic strategies to reduce tumor burden from the worldwide obesity epidemic.

## Results

**Dietary obesity induces hepatic CCRK to promote NASH.** To explore the role of CCRK in obesity-induced NASH, male C57Bl/6 mice were fed with diet of saturated fat and fructose for 22 weeks (Fig. 1a)[29,30]. The high-fat high-carbohydrate (HFHC)-fed mice exhibited significant increases in body weight ($p < 0.001$; Fig. 1b) and circulating levels of triglyceride ($p < 0.05$) and non-esterified fatty acid (NEFA; $p < 0.01$; Fig. 1c) when compared to mice fed with chow diet (CD). They also showed impaired glucose tolerance and insulin resistance as measured by IPGTT and IPITT, respectively ($p < 0.001$; Fig. 1d, e). Notably, CCRK was significantly up-regulated in the liver ($p < 0.01$; Fig. 1f), which was associated with hepatic steatosis, ballooning degeneration and spotty necrosis ($p < 0.01$; Fig. 1g–j). In contrast, the HFHC-fed female C57Bl/6 mice did not exhibit CCRK induction (Supplementary Fig. 1a), triglyceride/NEFA abnormalities (Supplementary Fig. 1b–c), or glucose intolerance (Supplementary Fig. 1d), although the weight gain was comparable to the male counterparts (Supplementary Fig. 1e).

We next investigated whether CCRK promotes NASH development in male mice via administration of lentivirus expressing short-hairpin RNA (shRNA) in the dietary obesity model (Fig. 1a). Lentiviral shRNA-mediated down-regulation of Ccrk (shCcrk) in the livers of the obese mice nearly restored the plasma triglyceride and NEFA concentrations to the basal levels ($p < 0.05$; Fig. 1c), and reversed the glucose intolerance and insulin resistance when compared to the control mice treated with lentivirus expressing non-specific sequence control (shCtrl; $p < 0.01$; Fig. 1d, e). Moreover, Ccrk knockdown significantly reduced hepatic lipid accumulation, ballooning degeneration, and spotty necrosis ($p < 0.05$; Fig. 1g–j), thus supporting a key role of CCRK in promoting NASH.

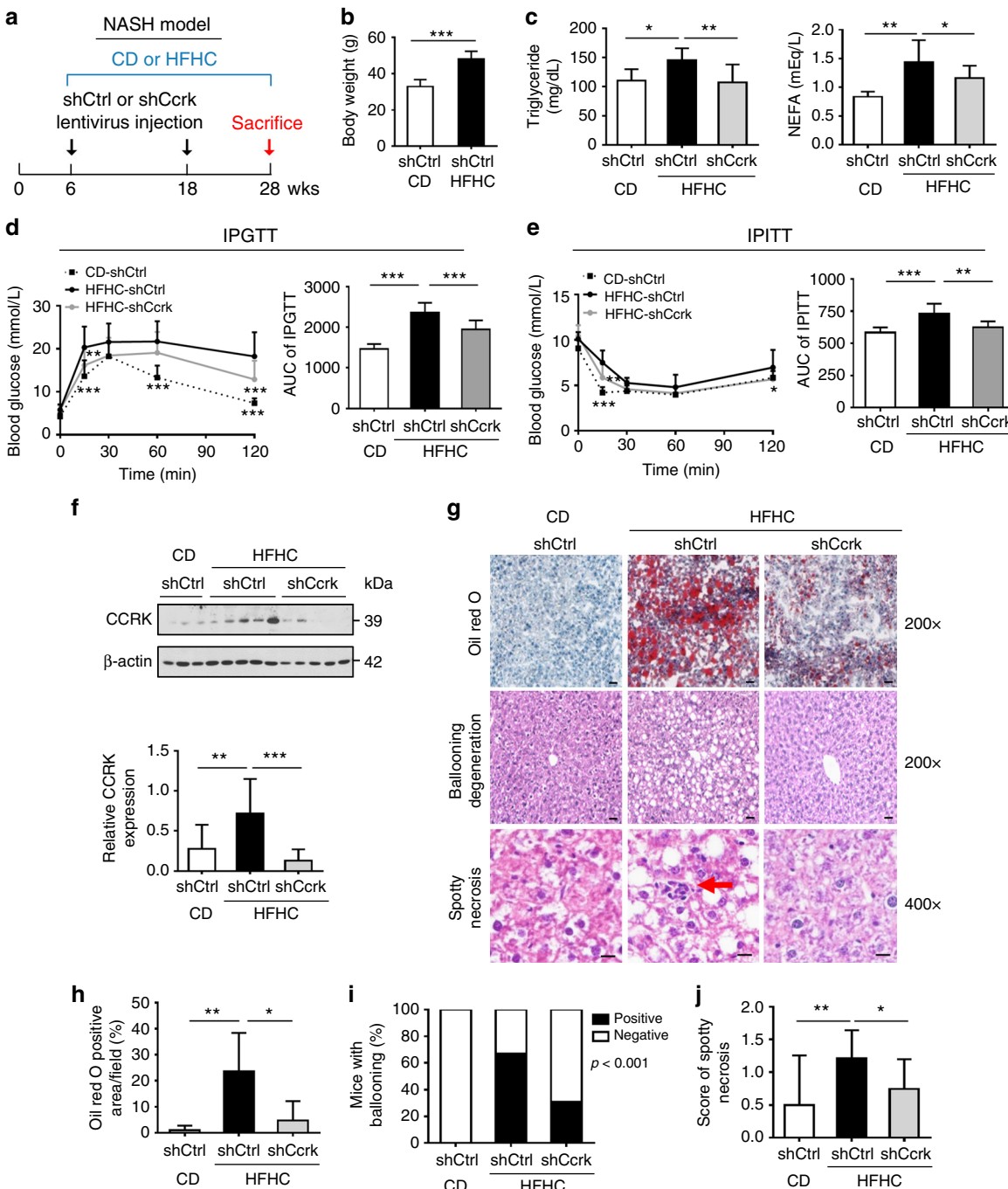

**Fig. 1** Dietary obesity-induced CCRK over-expression promotes lipid accumulation, glucose intolerance, and liver damages in male mice. **a** Schematic diagram of NASH mouse model with different diets (CD chow diet, HFHC high-fat high-carbohydrate) and lentivirus-mediated *Ccrk* knockdown (CD+shCtrl, *n*=8; HFHC+shCtrl, *n* = 15; HFHC+shCcrk, *n* = 15). **b** Body weight, **c** blood triglyceride and non-esterified fatty acid (NEFA) levels in mice were increased by HFHC at 28 weeks, which could be reduced by *Ccrk* knockdown. **d**, **e** CCRK impaired insulin sensitivity in mice. **d** Intraperitoneal glucose tolerance test (IPGTT) and **e** intraperitoneal insulin tolerance test (IPITT) were performed on CD-fed and HFHC-fed mice, blood glucose was measured at indicated time points after glucose or insulin injection (left), and area under the curve (AUC) is shown in a bar chart (right). **f** CCRK protein expression was induced by HFHC in mouse livers, which could be reduced by shRNA-mediated knockdown. Quantification of CCRK protein levels (relative to β-actin) is shown in a bar chart (bottom). **g** Representative pictures of Oil Red O, ballooning degeneration, and spotty necrosis of liver tissues in different groups (image magnification = ×200 or ×400, scale bar = 20 μm). **h**, **i** Quantifications of Oil Red O, ballooning degeneration and **j** scoring of spotty necrosis showed increased lipid accumulation, ballooning and spotty necrosis in HFHC-fed mice, which were reduced by *Ccrk* knockdown. Data are presented as mean ± SD. *$p < 0.05$; **$p < 0.01$; and ***$p < 0.001$ as calculated by unpaired two-tailed Student's *t*-test (**b**), one-way ANOVA followed by Bonferroni post-hoc test (**c–f**, **h**, **j**), two-way ANOVA followed by Bonferroni post-hoc tests (**d**, **e**), and Chi-square test (**i**)

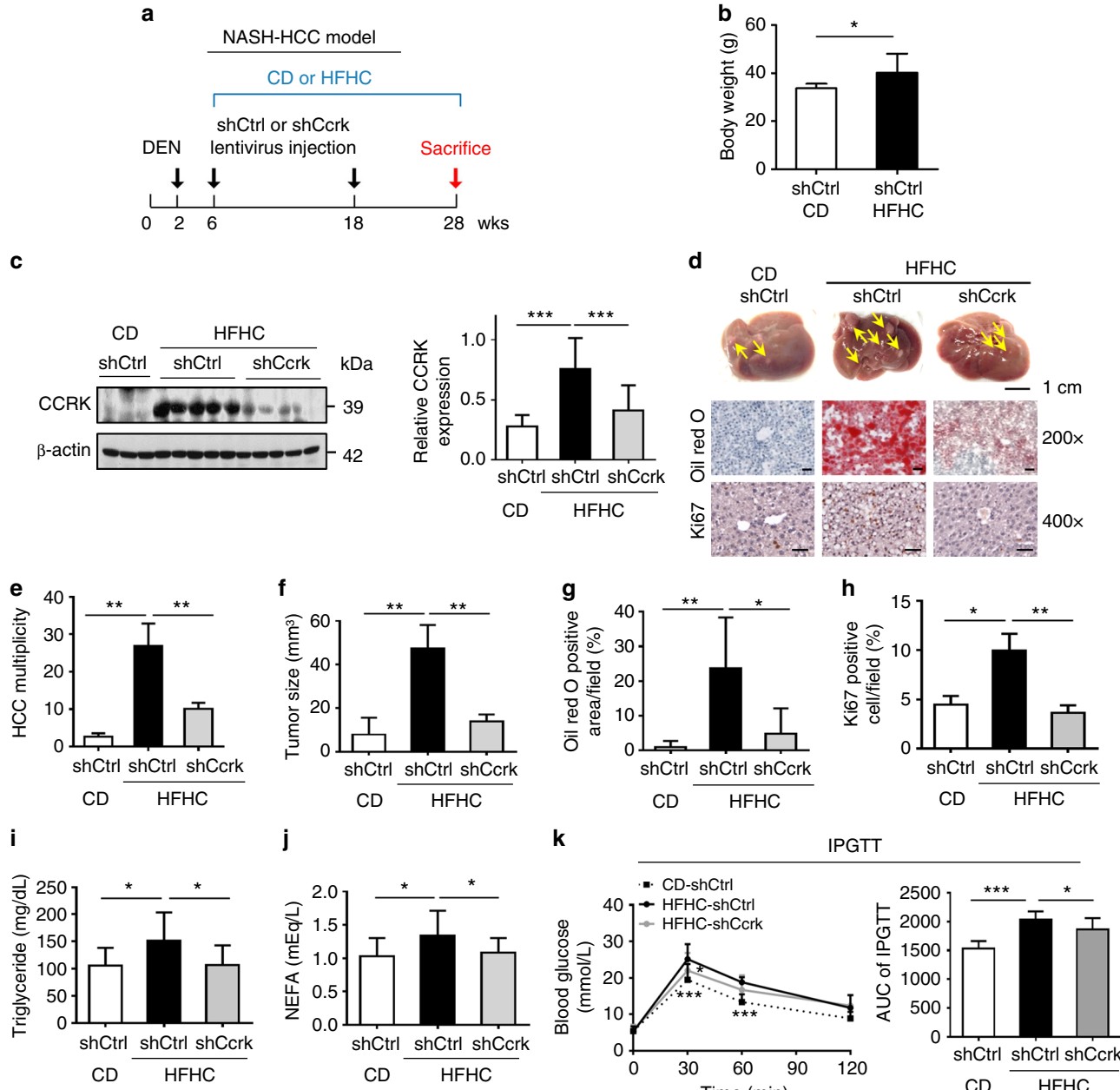

**Fig. 2** Knockdown of *Ccrk* improves glucose sensitivity, and reduces hepatic lipid accumulation and hepatocarcinogenesis. **a** Schematic diagram of NASH-HCC mouse model with carcinogen induction (DEN diethylnitrosamine), different diets, and lentivirus-mediated *Ccrk* knockdown (CD+shCtrl, $n = 8$; HFHC +shCtrl, $n = 15$; HFHC+shCcrk, $n = 15$). **b** Body weight of mice at 28 weeks. **c** CCRK protein expression in mouse livers were increased by HFHC at 28 weeks, which could be reduced by shRNA-mediated knockdown. Quantification of CCRK protein levels (relative to β-actin) is shown in a bar chart (right). **d** Representative pictures of dissected livers (scale bar = 1 cm), and Oil Red O and Ki67 staining of liver tissues in different groups (image magnification = ×200 or ×400, scale bar = 20 μm). **e** Tumor multiplicity and **f** tumor sizes were higher in the livers of HFHC-fed mice, which could be reduced by *Ccrk* knockdown. **g**, **h** Dietary obesity-induced CCRK increased lipid accumulation and cell proliferation, which could be abrogated by *Ccrk* knockdown as shown by Oil Red O staining and Ki67 staining in the livers. **i**, **j** Blood triglyceride and NEFA levels were elevated by dietary obesity-induced CCRK at 28 weeks, which could be reduced by *Ccrk* knockdown. **k** CCRK over-expression impaired insulin sensitivity in mice. IPGTT was performed on CD-fed and HFHC-fed mice, blood glucose was measured at indicated time points after glucose injection (left), and AUC is shown in a bar chart (right). Data are presented as mean ± SD. *$p < 0.05$; **$p < 0.01$; and ***$p < 0.001$ as calculated by unpaired two-tailed Student's *t*-test (**b**), one-way ANOVA followed by Bonferroni post-hoc test (**c**, **e–k**), and two-way ANOVA followed by Bonferroni post hoc test (**k**)

**CCRK mediates obesity-associated hepatocarcinogenesis**. To further investigate the oncogenic function of CCRK, we employed an obesity-associated HCC model[7,30] using HFHC-fed male mice with neonatal DEN treatment (Fig. 2a, b). In this model, hepatic CCRK was markedly up-regulated ($p < 0.001$; Fig. 2c), which was associated with much higher lipid accumulation and HCC

tumorigenicity when compared to the DEN-treated CD-fed mice (>10-fold; $p < 0.01$; Fig. 2d–g). Strikingly, when compared to shCtrl controls, *Ccrk* knockdown significantly reduced > 70% obesity-promoted tumor multiplicity and size in HFHC-fed mice ($p < 0.01$; Fig. 2e, f) but not in CD-fed mice (Supplementary Fig. 2a, b). Consistently, the livers of shCcrk-treated mice

exhibited significant reduction in lipid accumulation ($p < 0.05$) and hepatocellular proliferation ($p < 0.01$; Fig. 2g, h). The reduced tumorigenicity was also associated with significant restoration of normal circulating triglyceride/NEFA levels ($p < 0.05$; Fig. 2i, j) and glucose tolerance ($p < 0.05$; Fig. 2k). Taken together, our results suggest that CCRK plays a key role in promoting obesity-associated hepatocarcinogenesis.

**IL-6-mediated STAT3 and AR cooperatively induce CCRK**. We next investigated the molecular mechanism underlying the up-regulation of CCRK in obesity-associated hepatocarcinogenesis. As IL-6 represents a crucial HCC-promoting pro-inflammatory cytokine in obese state[7,9], we treated both HepG2 and Huh7 HCC cell lines with IL-6, and found increased phosphorylation of STAT3 at Tyr705 (p-STAT3$^{Tyr705}$) and elevated mRNA and protein expressions of CCRK, which were abrogated by short-interfering RNA (siRNA)-mediated knockdown of STAT3 (Fig. 3a, b). Luciferase reporter assay further demonstrated that IL-6 transcriptionally up-regulated *CCRK* in a STAT3-dependent manner (Fig. 3c). Consistently, we found that the serum IL-6 levels were significantly elevated ($p < 0.05$) and positively correlated with CCRK protein expression in both dietary obesity-induced NASH and HCC models ($p < 0.01$; Supplementary Fig. 2c, d).

Since *CCRK* is a direct transcriptional target of AR[24], we determined the potential interaction between IL-6/STAT3 and AR signaling in *CCRK* up-regulation. Knockdown of *AR* by siRNA abrogated IL-6-induced *CCRK* transcriptional activity and expression in HepG2 and Huh7 cells (Fig. 3a–c). Notably, deletion of the AR response element (ARE) in the *CCRK* promoter[24] completely abolished the *CCRK* transcriptional activation by IL-6 (Fig. 3c). Moreover, ectopic AR expression in STAT3 knockdown cells restored CCRK expression (Fig. 3d). Conversely, in CCRK-low-expressing LO2 immortalized hepatocytes and SK-Hep1 liver sinusoidal endothelial cells[31], ectopic expression of constitutive active STAT3 (STAT3C), but not dominant negative STAT3 (DN-STAT3) up-regulated AR and CCRK expression (Supplementary Fig. 3a), which could be abrogated by knockdown of *AR* (Supplementary Fig. 3b). Collectively, these data demonstrate a concerted action by IL-6/STAT3 and AR signaling to induce *CCRK* transcription and expression in liver and HCC cells, which was further supported by in vivo data from IL-6 neutralization experiment (Supplementary Fig. 3c, d).

**CCRK, STAT3, and AR form a positive feedback loop**. Given the pivotal roles of STAT3 and AR feedback circuits in HCC initiation[28,32], we speculated that CCRK feedback regulates STAT3 and AR signaling to form a positive loop. Ectopic CCRK expression in AR knockdown cells rescued p-STAT3$^{Tyr705}$ (Fig. 3e), while knockdown of *CCRK* in AR-over-expressing cells abrogated p-STAT3$^{Tyr705}$ (Supplementary Fig. 3e). Thus, AR activates STAT3 signaling in a CCRK-dependent manner. To further determine whether CCRK activates STAT3 signaling, we first performed CRISPR/Cas9-mediated *CCRK* deletion in Huh7 cells, which suppressed p-STAT3$^{Tyr705}$ and AR expression (Fig. 3f). Next, ectopic expression of wild-type (WT) CCRK, but not its kinase-defective (KD) mutant[25,26], increased p-STAT3$^{Tyr705}$ and AR expression in both LO2 and *CCRK* knockout (KO) Huh7 cells (Fig. 3f), suggesting a CCRK auto-regulatory loop.

We next investigated whether the auto-regulation occurs at the transcriptional level. Indeed, CCRK induced its own promoter activity, which was abolished by knockdown of either *STAT3* or *AR* in CCRK-expressing LO2 and SK-Hep1 cells (Fig. 3g and

Supplementary Fig. 3f). Deletion of ARE in the *CCRK* promoter also prevented the transcriptional activation, indicating that the auto-regulation was dependent on direct AR binding (Fig. 3g and Supplementary Fig. 3f). We next conducted co-immunoprecipitation to determine whether STAT3 physically interacts with AR for transcriptional regulation. WT but not KD CCRK induced a robust physical interaction between p-STAT3$^{Tyr705}$ and AR in LO2 and SK-Hep1 cells (Fig. 3h and Supplementary Fig. 3g). Furthermore, pairwise sequential ChIP assays showed that CCRK-induced co-localization of STAT3 and AR interaction in the ARE of its own promoter in both cell lines (Fig. 3i and Supplementary Fig. 3h). Taken together, these findings demonstrate that CCRK induces STAT3-AR physical interaction for transcriptional activation via promoter co-occupancy, thus forming an inflammatory-CCRK circuitry (Fig. 3j).

**CCRK activates TSC2/mTORC1 signaling through GSK3β**. We next elucidated the molecular pathways by which CCRK promotes NASH and HCC. Given the critical role of GSK3β, the direct substrate of CCRK[24] on mTOR signaling in obesity-associated HCC[20,22], it is conceivable that the CCRK/GSK3β cascade regulates mTORC1 signaling to promote obesity-associated hepatocarcinogenesis. We found that ectopic expression of CCRK in LO2 and *CCRK* KO Huh7 cells activated mTORC1 signaling as shown by increased levels of phosphorylated mTOR at Ser2448 (p-mTOR$^{Ser2448}$), 4E-BP1 at Thr37/46 (p-4E-BP1$^{Thr37/46}$), S6K at Thr389 (p-S6K$^{Thr389}$), and the mature form of SREBP1 (mSREBP1), which were impaired by suppression of GSK3β phosphorylation at Ser9 (p-GSK3β$^{Ser9}$) via over-expression of the constitutively-active S9A-GSK3β mutant (Fig. 4a). Moreover, knockdown of *CCRK* in HepG2 and Huh7 cells reduced mTOR phosphorylation and the downstream signaling cascades, which could be rescued by silencing of TSC2, an GSK3β effector on mTORC1 inhibition[33] (Fig. 4b). To further investigate whether this regulatory pathway is perturbed in vivo, we examined the expression of the signaling molecules in the dietary obesity models. In accordance with CCRK up-regulation, phosphorylation of GSK3β, mTOR, and the downstream signaling molecules were concordantly increased in the liver tissues of both NASH and NASH-HCC models (Fig. 4c, d). Moreover, down-regulation of *Ccrk* markedly suppressed the GSK3β/mTORC1/SREBP signaling (Fig. 4c, d) and deregulated multiple lipid metabolic pathways. For instance, *Ccrk* down-regulation resulted in significant reduction of *Srebf1* and its target genes *Acc1*, *Acl*, and *Fasn*, which are major drivers in de novo lipogenesis (Supplementary Fig. 2e). In parallel, *Fabp3* was down-regulated suggestive of reduced fatty acid uptake (Supplementary Fig. 2f), while *Abcg1* was up-regulated indicative of enhanced lipid secretion (Supplementary Fig. 2g). Intriguingly, *Cpt2* and *Ppara* were also down-regulated, denoting suppression of fatty acid beta-oxidation presumably due to decreased amount of lipids (Supplementary Fig. 2h). Of note, loss of *CPT2* was recently shown to promote HCC by protecting tumor cells from lipotoxicity[34] and chemotoxicity[35]. Collectively, these data suggest that CCRK-induced liver steatosis in obesity-associated hepatocarcinogenesis is a net result of increased lipid synthesis and uptake, and decreased lipid secretion.

**CCRK promotes lipid accumulation and insulin resistance via mTORC1**. Recent studies underline a strong link between mTORC1 signaling and NAFLD pathogenesis[36], which is associated with abnormalities in hepatic lipid and glucose metabolism as well as insulin resistance[37]. We thus hypothesized that over-

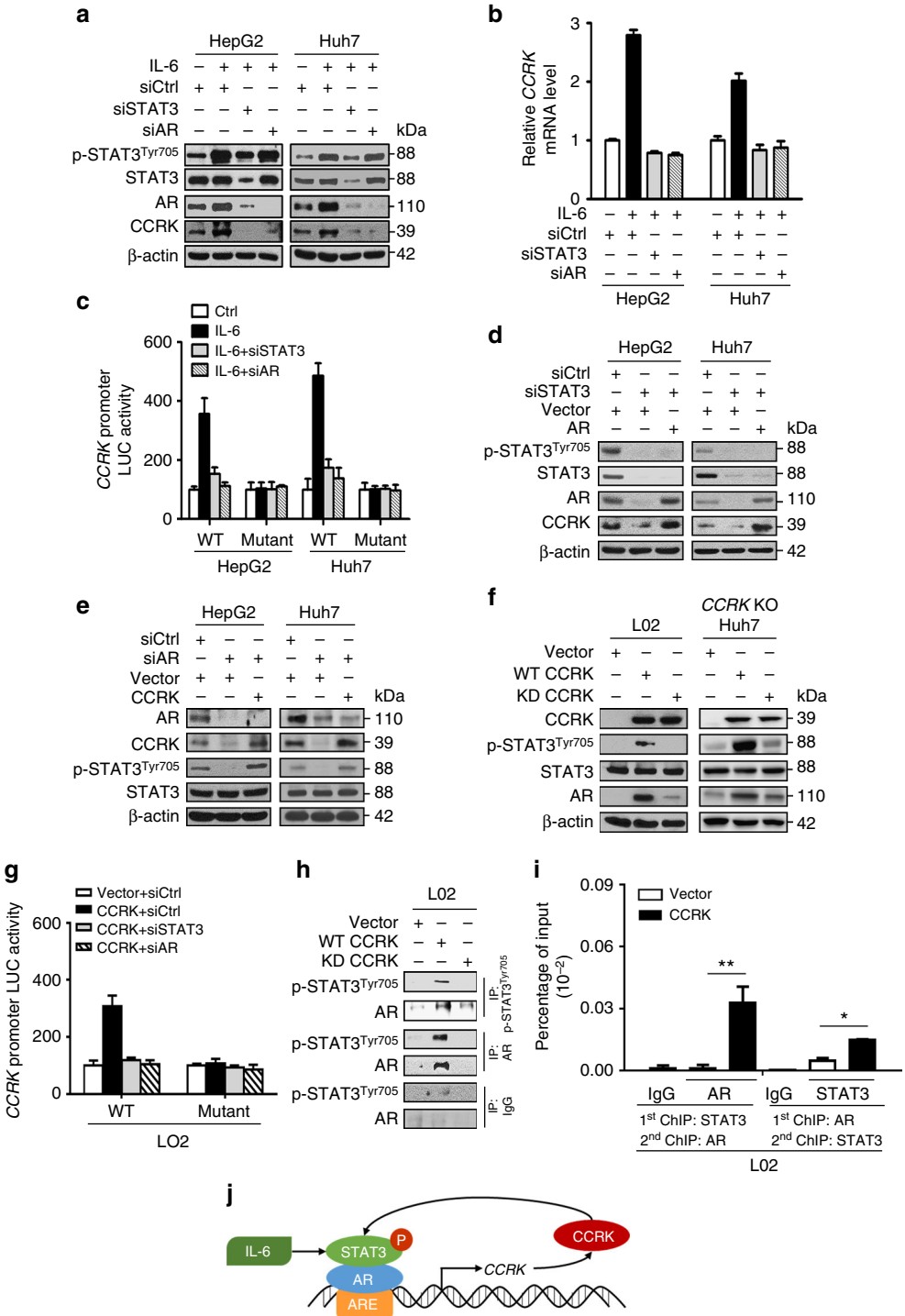

**Fig. 3** IL-6 activates STAT3 and AR signaling to stimulate CCRK expression. **a** Western blot and **b** qRT-PCR analysis of HepG2 and Huh7 cells treated with or without IL-6, combined with siRNA-mediated knockdown of *STAT3* or *AR*. **c** Luciferase reporter assay was used to measure *CCRK* promotor activity in HepG2 and Huh7 cells expressing either WT or AR response element (ARE)-deleted mutant *CCRK* promoter, treated with or without IL-6, combined with siRNA-mediated knockdown of *STAT3* or *AR*. **d** CCRK expression was blocked by knockdown of *STAT3*, but rescued by over-expression of AR. **e** STAT3 phosphorylation suppressed by *AR* knockdown could be rescued by over-expression of CCRK. **f** WT but not kinase-defective (KD) CCRK-induced STAT3 phosphorylation and AR expression. **g** CCRK induces its own promoter activity, which was abolished by deletion of ARE or knockdown of either *STAT3* or *AR* in CCRK-expressing cells. **h** Co-immunoprecipitation of p-STAT3$^{Tyr705}$ and AR in LO2 cells transfected with WT CCRK, but not in those transfected with empty vector or KD CCRK. IgG is a control for non-specific immunoprecipitation. **i** ChIP-re-ChIP assay demonstrated an increased co-occupancy of STAT3 and AR at the ARE of *CCRK* promotor in LO2 cells transfected with WT CCRK relative to those transfected with vector only. IgG is a control for non-specific immunoprecipitation. The antibodies used in the 1st and 2nd ChIP are as indicated. **j** Schematic representation of the IL-6/STAT3/AR/CCRK circuitry in hepatocarcinogenesis. Data are presented as mean ± SD. *$p < 0.05$ and **$p < 0.01$ as calculated by one-way ANOVA followed by Bonferroni post hoc test (**i**)

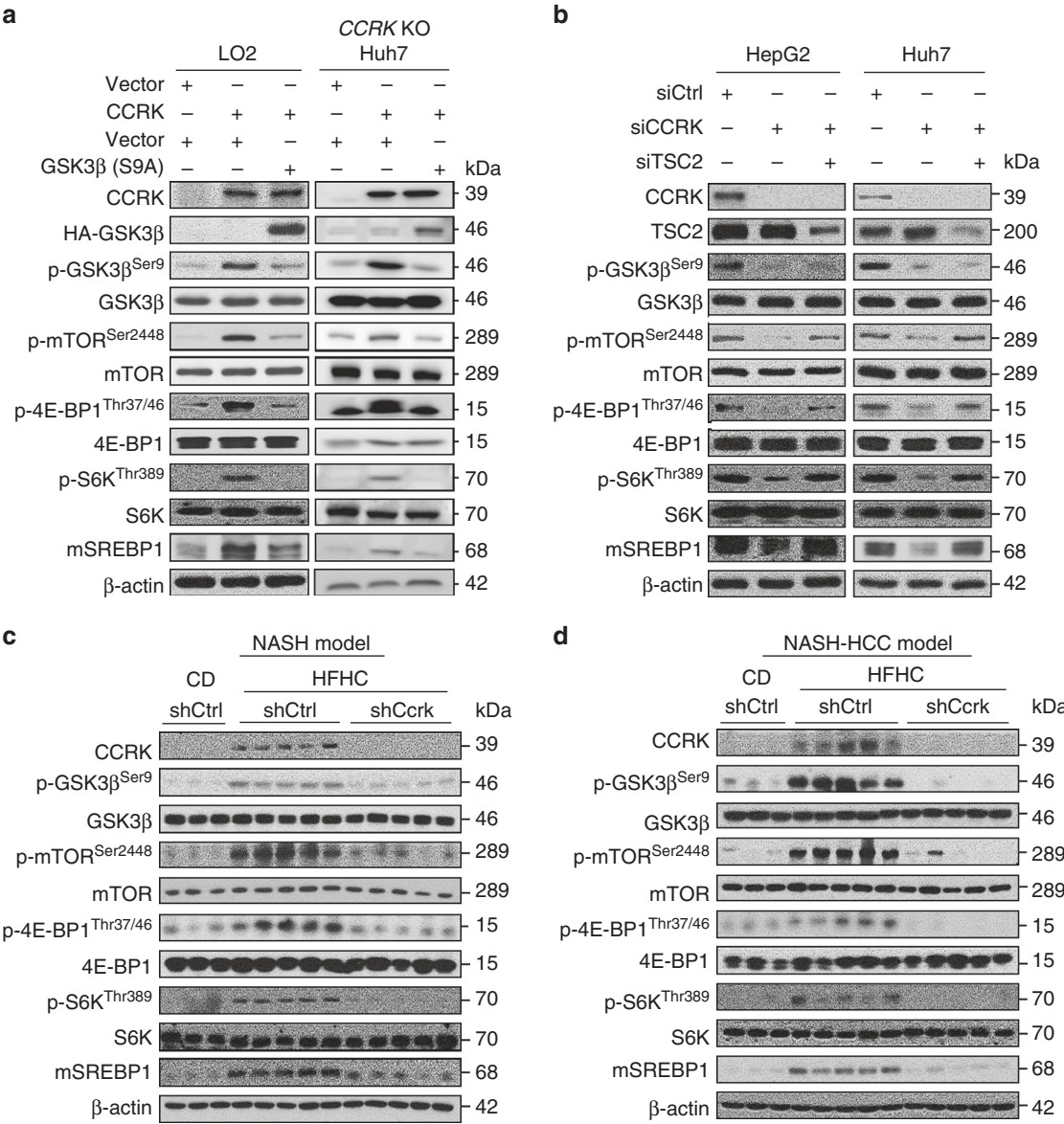

**Fig. 4** CCRK activates mTORC1 signaling through GSK3β/TSC2 cascade. **a** Ectopic CCRK expression in LO2 and *CCRK* KO Huh7 cells activates mTORC1 signaling, which was abrogated by suppression of GSK3β phosphorylation at Ser9 (p-GSK3β$^{Ser9}$) via over-expression of the constitutively-active S9A-GSK3β mutant. Western blot analysis was used to detect the protein expression of mTORC1 downstream molecules. **b** Knockdown of *CCRK* in HepG2 and Huh7 cells impaired the activation of mTORC1 signaling, which was rescued by the silencing of TSC2. **c**, **d** Dietary obesity-induced CCRK expression is responsible for the activation of mTORC1 signaling cascades in both **c** NASH and **d** NASH-HCC models, and such CCRK-dependent effects were abolished following knockdown of *Ccrk*. The establishment of NASH and NASH-HCC models are as described in Figs. 1a and 2a

expression of CCRK contributes to the pathophysiology of NAFLD via mTORC1 signaling. To test this, we first constructed a *CCRK*-inducible LO2 liver cell model (LO2-CCRK) in which doxycycline treatment readily induced CCRK, leading to increased p-GSK3β$^{Ser9}$, p-mTOR$^{Ser2448}$, p-4E-BP1$^{Thr37/46}$, p-S6K$^{Thr389}$, and mSREBP1 (Fig. 5a). Exposure of doxycycline-induced LO2-CCRK cells with fatty acid mixture of oleic and palmitic acids (2:1 ratio), which mimics the liver steatosis of NAFLD patients[38], significantly increased hepatocellular lipid accumulation compared to non-induced cells ($p < 0.001$; Fig. 5b, c). Notably, inhibition of the mTORC1 signaling by shRNA-mediated knockdown of the positive regulatory subunit *Raptor* (shRaptor) or, rapamycin treatment (Fig. 5a), abolished the lipid over-accumulation in CCRK-expressing liver cells ($p < 0.001$; Fig. 5b, c). In parallel, doxycycline-induced LO2-CCRK cells also exhibited significantly higher glucose uptake ($p < 0.001$;

Fig. 5d), which was completely abolished by mTORC1 inhibition ($p < 0.001$; Fig. 5d).

To determine whether CCRK promotes hepatic insulin resistance, we utilized *CCRK* KO Huh7 cells, which exhibited reduced GSK3β/mTORC1 signaling compared to the WT cells (Fig. 5e). We treated both WT and *CCRK* KO Huh7 cells with high dose of insulin for 24 h, followed by assessment of insulin sensitivity by phosphorylation of Akt at Ser473 (p-Akt$^{Ser473}$)[7,30,39]. High insulin exposure reduced subsequent insulin-stimulated p-Akt$^{Ser473}$ in Huh7 WT cells (Fig. 5f, lane 4 vs. lane 2), which could be restored in *CCRK* KO cells (Fig. 5f, lane 6 vs. lane 4). To investigate whether mTORC1 signaling mediates CCRK-promoted insulin resistance, we treated LO2-CCRK cells in hyperinsulinemia condition and found that induction of CCRK further impaired insulin sensitivity (Fig. 5g, lane 6 vs. lane 4). Notably, both genetic and pharmacological inhibition of

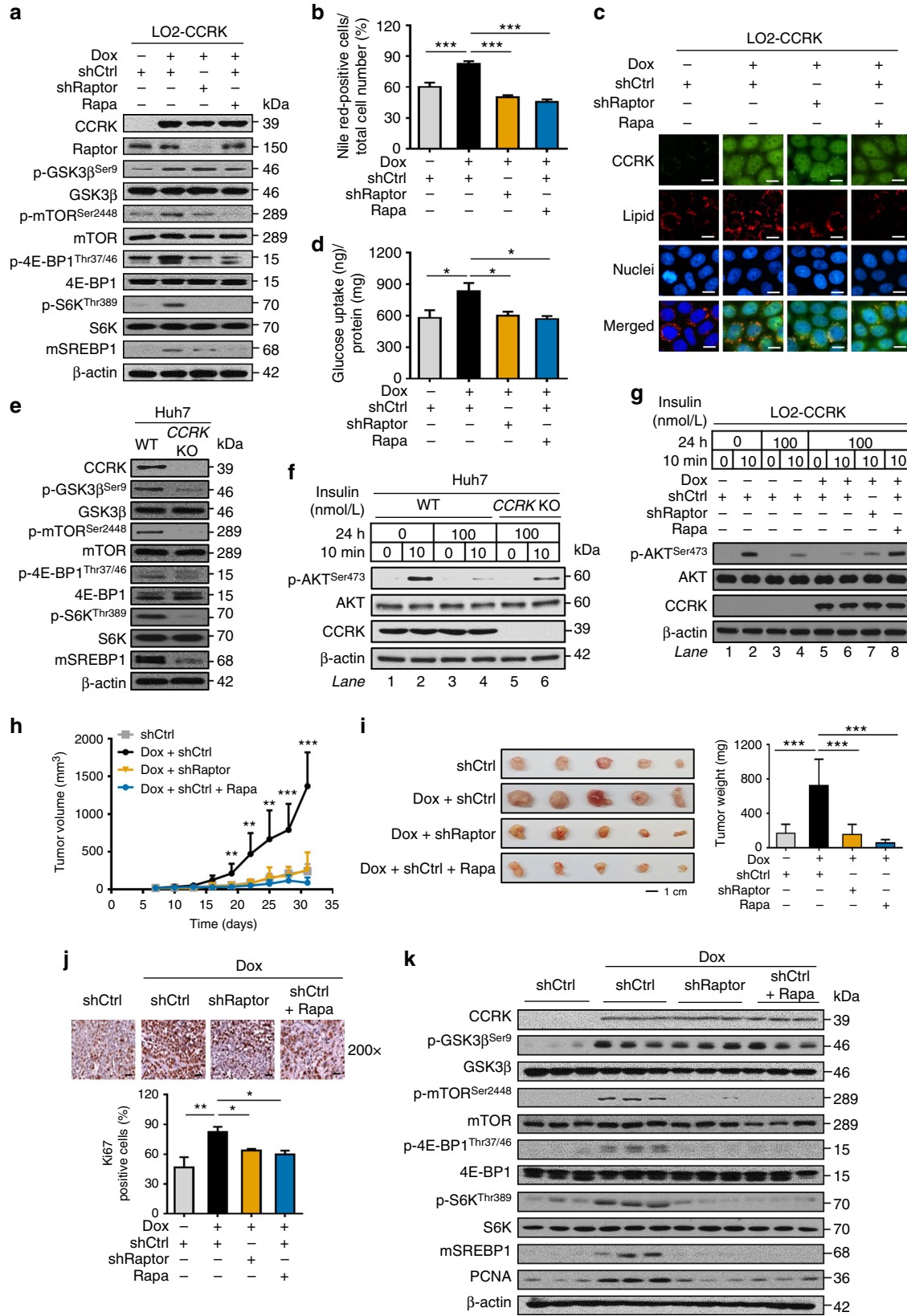

mTORC1 signaling rescued insulin sensitivity in CCRK-expressing cells (Fig. 5g, lane 7 and 8 vs. lane 6). Collectively, our findings suggest that hepatic CCRK fosters lipid accumulation, glucose uptake and insulin resistance through mTORC1 signaling.

**CCRK promotes mTORC1-dependent HCC tumorigenicity.** Next, we investigated whether mTORC1 signaling is required for CCRK-induced tumorigenicity by xenograft experiments. In contrast to non-induced LO2-CCRK cells, doxycycline-induced CCRK-expressing cells formed significantly larger and heavier

**Fig. 5** Hepatic CCRK promotes lipid accumulation, glucose uptake, insulin resistance, and tumorigenicity through mTORC1 activation. **a** Doxycycline (Dox)-induced expression of CCRK-activated mTORC1 signaling pathway in LO2-CCRK cells, which was abolished by inhibition of mTOR via *Raptor* knockdown or treatment with Rapamycin (Rapa). **b**, **c** Lipid accumulation (image magnification =×400, scale bar = 20 μm) as well as **d** glucose uptake were increased by CCRK-mediated mTOR activation, but were reduced by mTORC1 inhibition. **e** mTORC1 signaling was suppressed in *CCRK* KO Huh7 cell line. **f** CCRK impaired insulin sensitivity in Huh7 cells, which was restored by *CCRK* KO. Insulin sensitivity was assessed by p-Akt$^{Ser473}$ expression via Western blot analysis in cells treated with high dose of insulin followed by low dose of insulin stimulation. **g** The CCRK-induced insulin intolerance was restored by inhibition of mTORC1 using shRaptor or Rapamycin. **h**, **i** Mice injected with Dox-induced LO2-CCRK cells developed larger tumors (scale bar = 1 cm) compared to control mice and those treated with shRaptor or Rapamycin (*n* = 5 per group). **j**, **k** CCRK promoted tumorigenicity through mTORC1 activation. **j** The cell proliferation was assessed by Ki67 staining (scale bar = 20 μm). **k** The CCRK-activated mTORC1 signaling was detected by Western blot analysis. Data are presented as mean ± SD. *$p < 0.05$; **$p < 0.01$; and ***$p < 0.001$ as calculated by one-way ANOVA followed by Bonferroni post-hoc test (**b**, **d**, **i**, **j**), and two-way ANOVA followed by Bonferroni post-hoc test (**h**)

tumors ($p < 0.01$; Fig. 5h, i), whereas inhibition of mTORC1-signaling by either *Raptor* knockdown or Rapamycin treatment abrogated the induced tumorigenicity in CCRK-expressing cells ($p < 0.01$; Fig. 5h, i). Consistently, tumor nodules from doxycycline-induced LO2-CCRK cells showed significantly more proliferating cells when compared with non-induced, *Raptor*-ablated and Rapamycin-treated groups as assessed by Ki-67 staining ($p < 0.05$; Fig. 5j). Western blot analysis on the xenograft tumor tissues further confirmed that the reduced tumorigenicity was associated with the inactivation of GSK3β/mTORC1 signaling and its downstream molecules, and the cellular proliferation marker PCNA (Fig. 5k). Taken together, we demonstrate that CCRK promotes hepatic metabolic dysregulation and tumorigenicity in an mTORC1-dependent manner.

**Hepatic CCRK-mTORC1 signaling recruits MDSCs.** Since the liver environment plays an important role in tumor initiation and growth[40], we next employed an inducible and liver-specific *CCRK* transgenic (TG) mouse model[27] to investigate the pro-tumorigenic effect of CCRK via the hepatic immune microenvironment. At 10-day post-tamoxifen stimulation, CCRK expression was induced in the liver of *pTf-LSL-CCRK/+; Rosa26CreERt2/+*TG compared to *Rosa26CreERt2/+* control mice (Fig. 6a). Consistent with the in vitro findings, the hepatic p-GSK3β$^{Ser9}$ and p-mTOR$^{Ser2448}$ levels as well as the downstream p-4E-BP1$^{Thr37/46}$, p-S6K$^{Thr389}$, and mSREBP1 expressions were up-regulated (Fig. 6a), which were confirmed using TG mouse-derived primary hepatocytes (Fig. 6b). In accordance with the mTORC1/SREBP cascade activation, *CCRK* TG mice developed extensive liver steatosis as compared to the control mice when fed with HFHC diet ($p < 0.05$; Supplementary Fig. 4a–b).

We then assessed the tumor growth in control and *CCRK* TG mice using an orthotopic model via intrahepatic injection of syngeneic Hepa1–6 HCC cells (Fig. 6c). To exclude the influence of hepatoma-intrinsic CCRK on tumor growth and immunoregulatory function[27], we generated *Ccrk* KO Hepa1–6 cells via CRISPR/Cas9-mediated depletion (Fig. 6c). Notably, the hepatic tumorigenicity of *Ccrk* KO Hepa1–6 cells was significantly enhanced in *CCRK* TG compared to control mice ($p < 0.01$; Fig. 6d). Consistent with the oncogenic role of CCRK signaling via MDSC-mediated immunosuppression[27] (Supplementary Fig. 5a), we found that *CCRK* TG mice exhibited significantly higher liver-infiltrating level of CD11b$^+$Ly6C$^{Int}$Ly6G$^+$ polymorphonuclear MDSCs (PMN-MDSCs; $p < 0.05$; Fig. 6e), but not CD11b$^+$Ly6C$^+$Ly6G$^-$ monocytic MDSCs (M-MDSCs; Supplementary Fig. 5b). We next investigated whether the activated mTORC1 signaling promotes the enhanced tumorigenicity in *CCRK* TG mice. Lentiviral shRNA-mediated down-regulation of *Raptor* (shRaptor) in the livers of TG mice (Supplementary Fig. 5c) abolished tumor growth ($p < 0.05$; Fig. 6d), which was associated with significant reduction in PMN-MDSCs ($p < 0.05$; Fig. 6e). Moreover, the level of PMN-MDSCs, but not M-MDSCs,

in the liver positively correlated with tumor weight in the orthotopic HCC model ($p < 0.05$; Fig. 6f and Supplementary Fig. 5b). To determine its functional significance, we depleted PMN-MDSCs by intraperitoneal injection of anti-Ly6G antibody[41] after tumor cell implantation in *CCRK* TG mice (Supplementary Fig. 5d). Administration of anti-Ly6G antibody significantly suppressed CCRK-induced PMN-MDSC accumulation in liver ($p < 0.01$), leading to a borderline significant reduction in HCC tumorigenicity ($p = 0.0555$; Supplementary Fig. 5e, f).

We further validated these findings in the dietary obesity-HCC model (Fig. 2a). Consistently, the level of PMN-MDSCs, but not M-MDSCs, was also significantly increased in HFHC-fed mice ($p < 0.05$) but normalized to basal level by *Ccrk* knockdown ($p < 0.05$; Fig. 6g and Supplementary Fig. 5g). Overall, the level of PMN-MDSCs but not M-MDSCs significantly correlated with tumor multiplicity in the dietary obesity-HCC model ($p < 0.0001$; Fig. 6h and Supplementary Fig. 5g). To examine how hepatic CCRK expression induces PMN-MDSCs, we performed cytokine profiling and found a significant induction of *granulocyte-colony stimulating factor (G-csf)*[42] in the peri-tumoral liver tissues of *CCRK* TG mice ($p < 0.05$; Supplementary Fig. 5h). Consistently, the hepatic *G-csf* expression was significantly increased in HFHC-fed compared to CD-fed mice ($p < 0.05$), which was abrogated by *Ccrk* knockdown ($p < 0.01$; Fig. 6i). Using both ectopic expression and KO cell models, we further demonstrated that CCRK positively regulated *G-CSF* expression, which could be abolished by *Raptor* knockdown (Fig. 6j, k). Taken together, these results suggest that hepatic CCRK signaling recruits pro-tumorigenic PMN-MDSCs to liver microenvironment through mTORC1-dependent G-CSF expression.

**Co-activation of CCRK and mTORC1 in human NASH-HCCs.** To investigate the clinical relevance of our findings, the protein levels of CCRK, p-GSK3β$^{Ser9}$, GSK3β, p-mTOR$^{Ser2448}$, mTOR, p-4E-BP1$^{Thr37/46}$, 4E-BP1, p-S6K$^{Thr389}$, S6K, mSREBP1, G-CSF, p-STAT3$^{Tyr705}$, STAT3, and AR were examined by Western blot (Fig. 7a) in 23 pairs of human NASH-associated HCCs with neither viral hepatitis nor alcoholic liver disease (Supplementary Table 1). Compared with the paired non-tumor liver tissues, significant up-regulation of CCRK, p-GSK3β$^{Ser9}$/GSK3β, p-mTOR$^{Ser2448}$/mTOR, p-4E-BP1$^{Thr37/46}$/4E-BP1, p-S6K$^{Thr389}$/S6K, mSREBP1, G-CSF, p-STAT3$^{Tyr705}$/STAT3, and AR were detected in HCC tissues ($p < 0.05$; Fig. 7b). Quantitative RT-PCR also demonstrated significant elevation of the mRNA levels of *IL-6*, *CCRK*, and human MDSC markers *CD33* and *OLR1*[43] in the clinical specimens ($p < 0.05$; Supplementary Fig. 6a).

Association analysis further showed positive correlations between *IL-6* and *CCRK* ($r = 0.5403$), CCRK and *CD33* ($r = 0.4473$), CCRK and *OLR1* ($r = 0.4684$) ($p < 0.05$, Supplementary Fig. 6b), p-STAT3$^{Tyr705}$/STAT3 and CCRK ($r = 0.3691$), AR and CCRK ($r = 0.6507$), CCRK and p-GSK3β$^{Ser9}$/GSK3β ($r = 0.3066$),

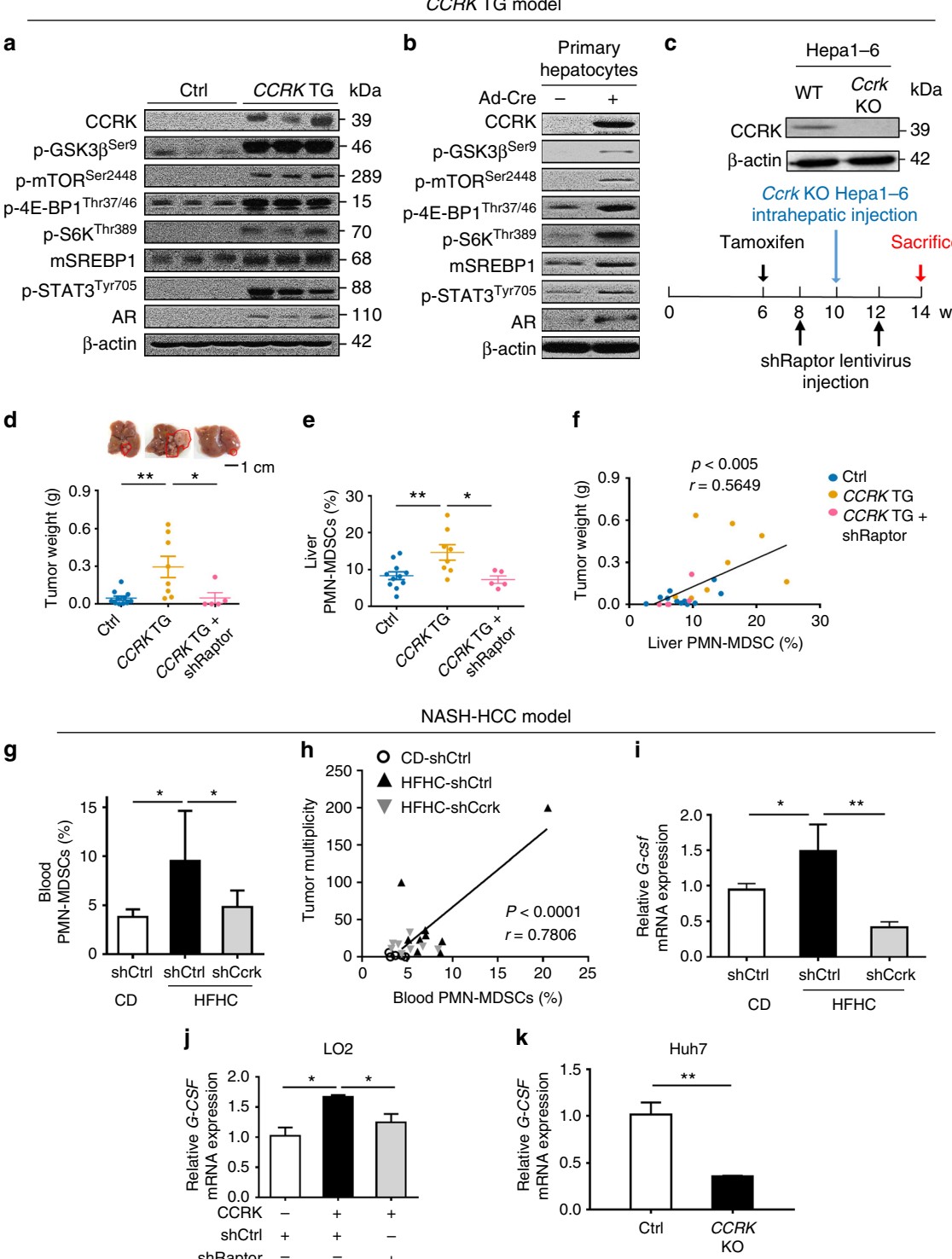

**Fig. 6** Hepatic CCRK-mTORC1 signaling recruits MDSCs to form a tumor-prone liver microenvironment. **a** mTORC1 signaling was activated in the livers of *CCRK* TG mice. The downstream molecules of mTORC1 signaling were detected by Western blot. **b** mTORC1 signaling components were activated in the primary hepatocytes isolated from *CCRK* TG mice. **c** Schematic diagram of *CCRK* TG mouse model with intrahepatic injection of *Ccrk* KO Hepa1–6 cells by CRISPR/Cas9, and lentiviral-mediated *Raptor* knockdown. **d** *CCRK* TG mice ($n = 8$) developed significantly larger tumors compared to controls ($n = 11$), which was abolished by down-regulation of *Raptor* (shRaptor; $n = 5$) (scale bar = 1 cm). **e** PMN-MDSCs were induced in *CCRK* TG mice, but were reduced after *Raptor* knockdown, and **f** PMN-MDSC levels positively correlated with tumor weight. **g** PMN-MDSCs were induced by HFHC diet, but were reduced by *Ccrk* knockdown in NASH-HCC model (CD+shCtrl, $n = 8$; HFHC+shCtrl, $n = 15$; HFHC+shCcrk, $n = 15$), wherein **h** PMN-MDSC levels positively correlated with tumor multiplicity. **i** *G-csf* mRNA levels were induced by HFHC diet, but were reduced by *Ccrk* knockdown in NASH-HCC model. **j** G-CSF mRNA level was augmented by CCRK over-expression in LO2 cells, which could be attenuated by *Raptor* knockdown. **k** G-CSF mRNA expression was down-regulated in *CCRK* KO Huh7 cells. Data are presented as mean ± SD. *$p < 0.05$ and **$p < 0.01$ as calculated by one-way ANOVA followed by Bonferroni post hoc test (**d**, **e**, **g**, **i**, **j**), Pearson correlation (**f**, **h**), and unpaired two-tailed Student's *t*-test (**k**)

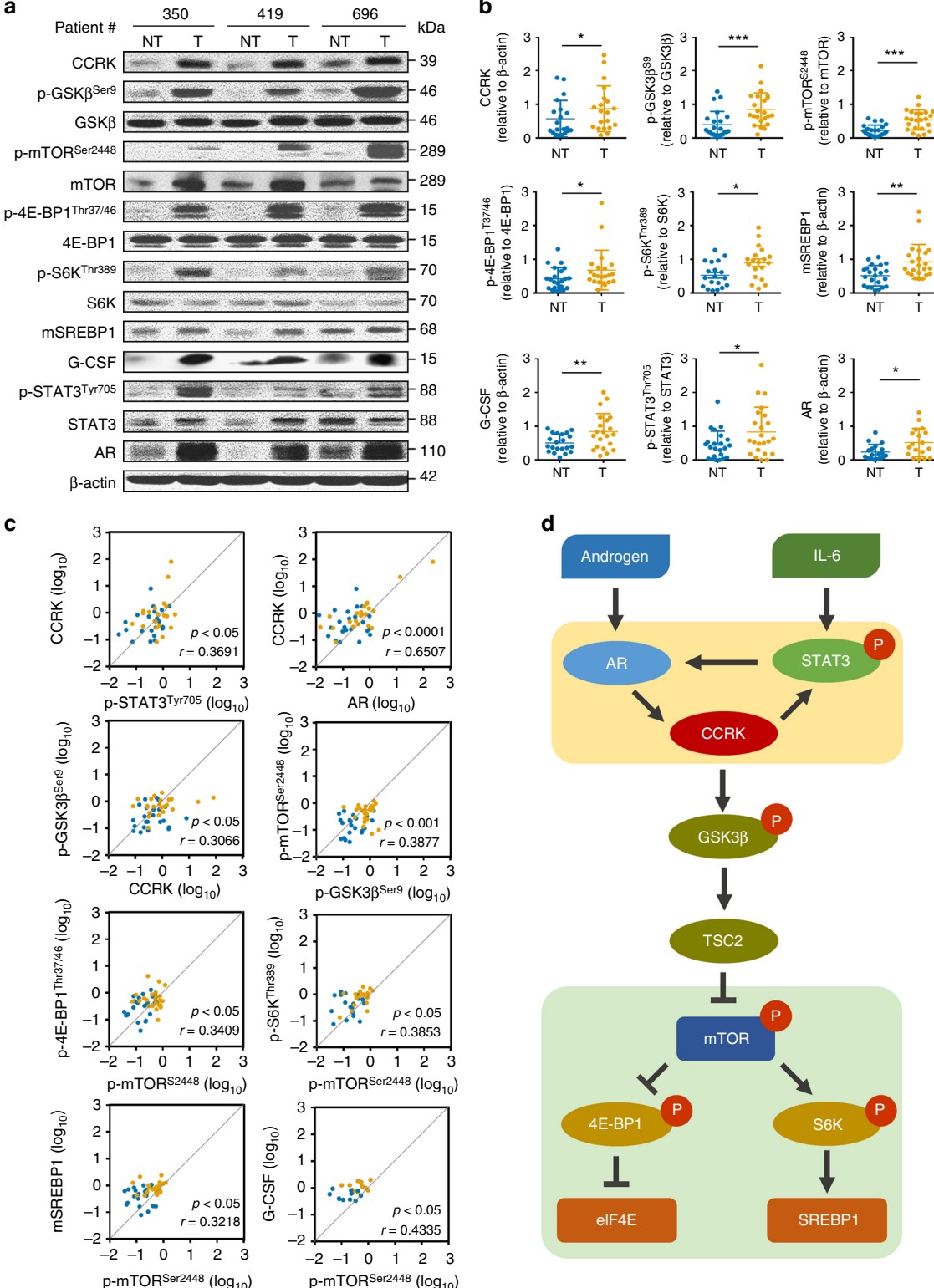

**Fig. 7** CCRK over-expression correlates with mTOR signaling activation in patients with NASH-associated HCCs. **a** Western blots showed the induced expressions of CCRK/mTOR signaling proteins in HCC tissues relative to matched non-tumor tissues (23 pairs), and representative results from three patients are shown. **b** The Western blot results of all patients from **a** were quantified and shown in dot plots, wherein the fold enrichments of target proteins relative to their corresponding controls are shown. **c** The positive correlations of STAT3/AR/CCRK/GSK3β/mTORC1 signaling components were confirmed in the patient specimens. **d** Schematic diagram showing the IL-6-triggered self-reinforcing circuitry of STAT3/AR/CCRK (highlighted in the yellow panel), which activates mTORC1 signaling (highlighted in the green panel) through GSK3β/TSC2 to promote NASH and HCC development. Data are presented as mean ± SD. *$p < 0.05$; **$p < 0.01$; ***$p < 0.001$ as calculated by unpaired two-tailed Student's $t$-test (**b**), and Pearson correlation (**c**)

p-GSK3β$^{Ser9}$/GSK3β and p-mTOR$^{Ser2448}$/mTOR ($r = 0.3877$), p-mTOR$^{Ser2448}$/mTOR and p-4E-BP1$^{Thr37/46}$/4E-BP1 ($r = 0.3409$), p-mTOR$^{Ser2448}$/mTOR and p-S6K$^{Thr389}$/S6K ($r = 0.3853$), p-mTOR$^{Ser2448}$/mTOR and mSREBP1 ($r = 0.3218$), and p-mTOR$^{Ser2448}$/mTOR and G-CSF ($r = 0.4335$) at protein levels ($p < 0.05$; Fig. 7c). Taken together, our findings in clinical HCC specimens consolidate our model of IL-6-triggered self-reinforcing STAT3/AR/CCRK circuitry, thereby activating mTORC1 signaling cascades to promote NASH-associated HCC development (Fig. 7d).

## Discussion

The rising prevalence of metabolic risk factors, especially obesity and diabetes, among HCC patients underscores the urgent need for novel therapeutic strategies[3,44,45]. Strikingly, the risk of obesity-associated HCC is even higher in males[1-3]. Here we show that hepatic CCRK cooperatively induced by the pro-inflammatory IL-6/STAT3 and AR signaling promotes HCC development by reprogramming lipid metabolism and immune microenvironment. Mechanistically, STAT3-AR co-binding stimulated by CCRK transcriptionally activates its own promoter, which in turn triggers the mTORC1/4E-BP1/S6K/SREBP1 cascades via GSK3β phosphorylation to augment hepatic lipid accumulation, glucose intolerance, insulin resistance, and tumorigenicity. Moreover, over-expression of CCRK reshapes the liver microenvironment by up-regulating mTORC1-dependent G-CSF expression to recruit pro-tumorigenic PMN-MDSCs, resulting in immune evasion. As we also showed concordant STAT3-AR-CCRK-mTORC1 signaling activation in patients with NASH-associated HCC, our findings unveil a central role of CCRK as a molecular hub of multiple metabolic and immuno-suppressive cascades, and offer a novel therapeutic target for HCC intervention in the era of obesity epidemic.

Mounting evidence suggests that obesity is associated with chronic low-grade systemic inflammation leading to metabolic syndrome and insulin resistance[46]. The pro-inflammatory IL-6 levels are higher in mouse and human NAFLDs[10], and a positive correlation between hepatic IL-6 expression and disease severity was evident in patients with NASH[47]. Naugler et al. have previously shown that estrogen effectively suppressed IL-6 expression in female mice via MYD88/NF-κB signaling, and thereby reduced liver damages and HCC multiplicity[48]. In this murine HCC model, the IL-6 response was transient that occurred only during the first 48 h after DEN stimulation, whereas hepatocarcinogenesis was a long process that took almost 8 months[48]. These results implied that a sustainable oncogenic signal is required but not clearly defined. In this study, we addressed this enigma by showing in the male NAFLD context (Figs. 1 and 2) that IL-6 can induce a self-reinforcing STAT3/AR/CCRK oncogenic circuitry, in which CCRK facilitates the physical interaction of phosphorylated STAT3 and AR and their co-occupancy at CCRK promoter for sustained transcriptional activation (Fig. 3). As we also observed CCRK up-regulation in male HFHC-fed mice (Supplementary Fig. 1a) and NAFLD patients (Supplementary Fig. 6c), the IL-6/STAT3 and androgen/AR crosstalk in CCRK activation may explain why obese men are at greater risk of developing HCC.

The diverse functional effects of mTOR pathways in liver metabolism and hepatocarcinogenesis are well-established[15-19], however, the upstream drivers have not been fully defined. We found that in both NASH and HCC murine models, mTOR signaling was consistently activated and driven by CCRK (Fig. 4). Mechanistically, CCRK phospho-inactivates GSK3β, thereby subverts the mTOR inhibitor TSC2 to up-regulate p-4E-BP1, p-S6K, and mSREBP1 levels. As a negative regulator of a key

rate-limiting initiation factor eIF4E, 4E-BP1 phosphorylation by mTORC1 leads to the dissociation of 4E-BP1 from eIF4E, allowing the formation of cap-dependent translation initiation complex at 5′ end of mRNAs[14]. Once phosphorylated and activated by mTORC1, p-S6K1/2 is another major effector that regulates RPS6, as well as other regulators of translation initiation[15]. Notably, both of these critical pathways for protein synthesis have been shown to play indispensable roles for mTORC1-dependent hepatocarcinogenesis[49]. Additionally, increased de novo lipogenesis via the mTORC1/S6K1/SREBP1 axis has pathogenic and prognostic significance in HCC[15,17,50]. A recent study has further shown that either genetic or pharmacological inhibition of the SREBP pathway dramatically reduced HCC progression[51]. Importantly, we demonstrated that in vivo ablation of Ccrk simultaneously circumvents these metabolic and lipogenic mTORC1-dependent cascades to diminish hepatic lipid accumulation, glucose intolerance, insulin resistance, and HCC tumorigenicity (Fig. 5). Notably, CCRK also controlled the β-catenin-AR regulatory loop (Supplementary Fig. 7) as previously shown in HBV-associated hepatocarcinogenesis[26], which may contribute to the lipogenic tumor phenotype[18] in the HFHC diet model. Overall, these mechanistic and functional data highlight CCRK as a crucial mTORC1 regulator in the development of NASH and NASH-associated HCC.

Both cellular and non-cellular components of the tumor microenvironment determine tumor development and progression, as well as anti-tumor immunity and response to cancer therapy[40]. Interestingly, in addition to the tumor-intrinsic roles, CCRK also exerts tumor-extrinsic functions in the liver microenvironment[52]. Using a CCRK TG model, we showed that CCRK effectively induced mTORC1-dependent G-csf expression to recruit PMN-MDSCs to the liver, leading to enhanced tumorigenicity potentially via immune escape (Fig. 6). In the dietary obesity HCC model, we further found that the PMN-MDSC level (Fig. 6g) and hepatic p-p65$^{ser536}$ expression (Supplementary Fig. 7) were positively regulated by CCRK. As PMN-MDSCs were also increased in male CCRK TG mice under HFHC diet (Supplementary Fig. 4c), our data suggest that the obesity-triggered AR/CCRK signaling may establish an immunosuppressive barrier through induction of MDSC expansion and recruitment via NF-κB/IL-6[27] and mTORC1/G-CSF[42] pathways, respectively. Ma et al. have recently depicted the role of lipid dysregulation on adaptive immune responses, especially the selective loss of intrahepatic CD4$^+$ T lymphocytes in NAFLD-promoted HCC[53]. Thus, whether and how CCRK-driven signaling network modulates and impairs T cell-mediated tumor immune surveillance warrant further investigation.

Our findings of both intrinsic effects on hepatocytes and hepatocyte-extrinsic functions in the liver microenvironment highlight the therapeutic potential of targeting CCRK in HCC. Pharmacological inhibition of the aberrantly activated mTOR in HCC has been proposed, but the in vivo effects have not been clearly elucidated[54]. Rapamycin was even shown to produce unwanted adverse effects in HCC pre-clinical models[55]. As an upstream driver of mTOR pathway, Ccrk inhibition is sufficient to reverse the metabolic and oncogenic phenotypes in multiple NASH and HCC models. Moreover, a recent study showed that co-blockade of CCRK improved immune-checkpoint therapy for large hepatoma eradication[27,56]. There are currently a number of pan-CDK inhibitors that display binding to CCRK, but CCRK-specific inhibitor has yet been developed[56]. Given the functional and clinical significance of CCRK in both viral[26] and NASH-related HCCs (Fig. 7 and Supplementary Fig. 6d), the development of CCRK-targeted agents and characterizations of their effects in combination immunotherapy will ultimately lead to more effective treatment for patients with HCC.

## Methods

**Patient specimens**. Paired tumor/non-tumor tissues from 23 patients with NASH-associated HCCs who underwent hepatectomy (Supplementary Table 1), and liver biopsy tissues from 23 NAFLD patients (Supplementary Table 2), were collected at the Prince of Wales Hospital (Hong Kong) for analyses. All HCC patients reported a history of metabolic syndrome involving diabetes, hypertension, dyslipidemia, and hepatic steatosis. HCC patients with chronic hepatitis B and C, or record of excessive alcohol intake were excluded from this cohort. Informed consent was obtained from all human subjects, and the study was approved by the Joint CUHK-NTEC Clinical Research Ethics Committee. Liver tissues were graded by the NASH Clinical Research Network scoring system[57].

**Cell culture and transfection**. HepG2 (ATCC, HB-8065), Huh7 (JCRB, 0403), LO2 (Cellosaurus, CVCL_6926), and SK-Hep1 (ATCC, HTB-52) human cell lines and Hepa1–6 (ATCC, CRL-1830) mouse cell line were cultured in DMEM supplemented with 10% FBS (Gibco). Cell transfection was conducted using FuGENE® HD Transfection Reagent (Promega) or Lipofectamine 2000 reagent (Invitrogen) according to the manufacturer's instructions.

**Plasmid, RNA interference, and transfection**. For plasmid transfection, constitutively active and Y705F-DN-STAT3 (STAT3C and DN-STAT3) were a gift from Linzhao Cheng (Addgene plasmid #24983, #24984)[58]. CCRK-expressing vector was provided by Dr. Marie Lin (Chinese University of Hong Kong). A (kinase-defective) KD mutant of CCRK in which the T-loop threonine-161 was substituted with alanine was constructed as previously described[59]. A dominant-inhibitory phosphorylation-defective GSK-3β mutant in which the serine-9 was replaced with alanine (GSK-3β(S9A)) was a gift from Brendan Manning (Addgene plasmid #14128). Raptor_shRNA (shRaptor) was a gift from David Sabatini (Addgene plasmid #1858)[60]. The non-silencing shRNAmir control (shCtrl, RHS4346) was purchased from Thermo Fisher. shRNA vectors targeting mouse Ccrk (shCcrk) was constructed as previously described[26]. Cell transfection was conducted using FuGENE® HD Transfection Reagent (Promega) according to the manufacturer's instructions. Briefly, plasmids were mixed with transfection reagent at the 1–2 μg (plasmids): 3–6 μl (reagent) ratio for 15 min at room temperature and then evenly dropped onto the cells. The transfected cells were cultured in the incubator till needed.

Sense and antisense strands of siRNA oligonucleotides were synthesized and purified by Techdragon Company. The lyophilized powder of siRNAs was dissolved in DNase-/RNase-free water and aliquoted for storage at −80 °C. siRNAs specifically targeting human CCRK, AR, STAT3, and TSC2 were designed. A scramble siRNA (siCtrl) was synthesized as a negative control, which does not target any human gene mRNAs. Cells were transfected with siRNAs using Lipofectamine 2000 reagent (Invitrogen) according to the manufacturer's protocol. In brief, at 24 h before transfection, cells were seeded (in antibiotic-free media) at 70–90% confluency. siRNAs and Lipofectamine 2000 were individually incubated with serum-free MEM medium (Gibco) for 5 min at room temperature, then they were mixed together and incubated at room temperature for 20 min. The mixtures were then pipetted onto the cells evenly, and the cells were cultured in the incubator until indicated time point.

**Construction of inducible and stable cell lines**. Generation of the doxycycline-inducible stable cell line was performed as described[61]. In brief, CCRK cDNA was amplified by PCR and used to replace the PAF gene of the vector PB-T-PAF to form the new construct PB-T-CCRK. LO2 cells were seeded and grown in T75 flasks overnight to reach 60–70% confluency. On the following day, cells were co-transfected with pCyL43, PB-RN together with or without PB-T-CCRK using Lipofectamine 2000 following the manufacturer's instructions. Two days after transfection, cells were subjected to dual drug selection, i.e. 1 μg/ml puromycin (Gibco) and 800 μg/ml G418 (Roche), until drug-resistant colonies formed and stabilized.

For constructing shRaptor and shControl (shCtrl) stable cell lines, doxycycline-inducible CCRK-expressing LO2 cells (LO2-CCR) were seeded and grown in six-well plates overnight to reach 50–70% cell confluency. Cells were transfected with shRaptor or shCtrl plasmid using FuGENE® HD Transfection Reagent and cultured for 48 h. After that, transfected cells were detached from the plates with 0.25% trypsin-EDTA, and transferred to a 100-mm tissue culture plate and maintained in 400 μg/ml hygromycin (Life Technologies Corporation) for 4 weeks until resistant colonies formed. Then, the stable cells (shRaptor or shCtrl-LO2) were isolated and transferred into new dishes and cultured in antibiotics-containing medium for further experimental use.

CCRK KO Huh7 cell line was constructed as previously described[62]. Briefly, human CCRK-targeted sgRNA was designed online (http://crispr.cos.uni-heidelberg.de) and cloned into the vector lentiCRISPR v2 (Addgene Plasmid #52961). This construct was confirmed by sequencing. Next, Huh7 cells were transfected with this construct or control vector. Two days after transfection, puromycin (1 μg/ml) was used to select transfected colonies. Approximately 2 weeks later, individual cells were plated in 96-well plates and cultured for 4 weeks. CCRK KO Huh7 cells were confirmed by sequencing and Western blot.

**Quantitative reverse transcription PCR (qRT-PCR)**. Total RNAs were extracted using TRIZOL Reagent (Invitrogen). In each reaction 500 ng of total RNA was used to generate cDNA. RNA samples were mixed with DNase (Invitrogen, CA, USA) to get rid of unwanted genomic DNA and incubated with PrimeScript RT Master Mix (Takara). The mixture was incubated at 37 °C, 15 min; 85 °C, 5 s. The cDNA was diluted in 150 μl DNase/RNase-free water and stored at −20 °C until use. Aliquots (3 μl each) of cDNA were amplified using Power SYBR Green PCR Master Mix (Takara) and ViiTM7 Real-Time PCR System (Applied Biosystems). Each reaction was performed in triplicate and GAPDH was used as an internal control. The primers are listed in Supplementary Table 3.

**Western blot**. The primary antibodies for western blotting used in this study are CCRK (71485, Abcam, 1:1000), pSTAT3 (9131, Cell Signaling Technology, 1:1000), STAT3 (D3Z2G, 12640, Cell Signaling Technology, 1:1000), p-AR (156C135.2, sc-52894, Santa Cruz Biotechnology, 1:1000), AR (PG21, 06–680, EMD millipore, 1:1000), HA tag (C29F4, 3724, Cell Signaling Technology, 1:1000), p-GSK3β (9336, Cell Signaling Technology, 1:1000), GSK3β (D5C5Z, 12456, Cell Signaling Technology, 1:1000), p-mTOR (2971, Cell Signaling Technology, 1:1000), mTOR (2972, Cell Signaling Technology, 1:1000), p4E-BP1 (236B4, 2855, Cell Signaling Technology, 1:1000), 4E-BP1 (53H11, 9644, Cell Signaling Technology, 1:1000), p-S6K (D5U1O, 9205, Cell Signaling Technology, 1:1000), S6K (49D7, 2708, Cell Signaling Technology, 1:1000), mSREBP1 (2A4, sc-13551, Santa Cruz, 1:1000), pAKT (D9E, 4060, Cell Signaling Technology, 1:1000), AKT (11E7, 4685, Cell Signaling Technology, 1:1000), PCNA (PC10, MS106-P1, NeoMarkers, 1:1000), Raptor (24C12, 2280, Cell Signaling Technology, 1:1000), G-CSF (5D7, ab9818, Abcam, 1:1000), active β-catenin (D2U8Y, 19807, Cell Signaling Technology, 1:1000), β-catenin (D10A8, 8480, Cell Signaling Technology, 1:1000), p-P65 (93H1, 3033, Cell Signaling Technology, 1:1000), P65 (D14E12, 8242, Cell Signaling Technology, 1:1000), and β-actin (8H10D10, Cell Signaling Technology, 1:10,000). Protein lysates from cell lines and tissues were prepared using lysis buffer (50 mM Tris–HCl, pH 7.5, 150 mM NaCl, 1% NP-40, 0.5% Na-deoxycholate) and T-PER Tissue Protein Extraction Reagent (Thermo Scientific) supplemented with protease inhibitor cocktail, respectively. Protein concentration was determined by the Bradford method (Bio-Rad Laboratories). 10–100 μg of protein was separated by 6–12% sodium dodecyl sulfate polyacrylamide gel electrophoresis and electro-blotted onto equilibrated nitrocellulose membrane (Bio-Rad Laboratories). The indicated primary antibodies were incubated at 4 °C overnight followed by secondary antibodies for 2 h at room temperature. Then, the antibody–antigen complexes were detected with enhanced chemiluminescence (ECL, GE Healthcare Life Sciences) and exposed to X-ray film (Fuji). β-actin was used as loading control. Signals were quantified by Image J software and defined as the ratio of target protein to β-actin. Uncropped Western blots are shown in Supplementary Fig. 8.

**Co-immunoprecipitation**. Cells were collected and lysed in 1 ml co-IP lysis buffer (20 mM Tris, pH 7.5, 15 0 mM NaCl, 1.0% Triton X-100, 1 mM EDTA, and protease inhibitor cocktail) on ice. The lysates were centrifuged at 12,000×g, 15 min, 4 °C and immunoprecipitated with IgG, anti-p-STAT3^Tyr705 or anti-AR antibodies at 4 °C overnight. On the following day, 20 μl of protein A/G beads (Santa Cruz) were added into the protein–antibody complex and incubated for 6 h at 4 °C. The complexes were then washed with Co-IP lysis buffer four times for 10 min each at 4 °C and collected by centrifuging at 2500 rpm, 5 min, 4 °C. Proteins were detached from beads by denaturing with 5 × protein loading buffer at 100 °C for 10 min. Protein interaction was then detected by western blot using indicated antibodies.

**Chromatin immunoprecipitation (ChIP)-re-ChIP assay**. ChIP-re-ChIP assay was performed as previously described[30]. At 48 h post-transfection of CCRK and control plasmids, cells grown at the confluency of 80% in 150-mm dishes were cross-linked with 1% formaldehyde for 10 min at room temperature on the shaker, followed by adding 1 M glycine to a final concentration of 125 mM for 2 min to quench formaldehyde. Next, cells were washed twice using PBS at 4 °C and harvested by scraping. A total of 1 × 10^8 cells were collected at 700×g, 5 min, 4 °C and lysed with 10 ml cold lysis buffer I (50 mM HEPES–KOH, pH 8.0, 1 mM EDTA, 140 mM NaCl, 10% glycerol, 0.5% NP-40, 0.25% Triton X-100, supplemented with protease inhibitors), and rotated at 4 °C for 10 min. Then, the cell debris was harvested through centrifugation at 1350×g, 5 min, 4 °C, and then resuspended in 10 ml cold lysis buffer II (10 mM Tris–HCl, pH 8.0, 200 mM NaCl, 1 mM EDTA, 0.5 mM EGTA, supplemented with protease inhibitors) and rotated for 10 min at 4 °C. The nuclear pellets were collected by centrifugation at 1350×g, 5 min, 4 °C and resuspended with 3.6 ml lysis buffer III (10 mM Tris–HCl, pH 8.0, 100 mM NaCl, 1 mM EDTA, 0.5 mM EGTA, 0.1% Na-Deoxycholate, 0.5% N-Lauroylsarcosine, supplemented with protease inhibitors). DNA fragmentation was achieved using a Bioruptor ultrasonicator (Diagenode). 50 μl of cell lysate was saved as a reference sample. 5 μg of STAT3 antibody (Cell signaling) or AR antibody (Millipore) attached to 100 μl Dynabeads Protein G (Invitrogen) were used for immunoprecipitation of protein–DNA complexes overnight. For Re-ChIP assay, the beads were eluted in 10 mM DTT and mixed with Re-ChIP dilution buffer (50 mM Tris/HCl at pH 7.4, 150 mM NaCl, 1% Triton, 2 mM EDTA, and protease inhibitors) and subjected to a second ChIP. 5 μg of AR antibody

(Millipore), STAT3 antibody (Cell signaling), or IgG antibodies (Santa Cruz) attached to 100 μl Dynabeads Protein G (Invitrogen), which was subjected to re-ChIP assay to bind with protein–DNA complexes for immunoprecipitation overnight. The IP DNA or input DNA was subjected to elution, reverse crosslink and purification. PCR primers targeting a region within 150 bp of the putative-binding site were designed to test the IP and input DNA. Equal amounts of IP and diluted input DNA were used as templates for quantitative PCR by Power SYBR Green-based detection (Applied Biosystems).

**Luciferase reporter assay.** The WT and androgen-responsive element (ARE)-deleted CCRK promoter luciferase reporters were constructed as previously described[24]. For IL-6-induced CCRK transcription, cells were transiently trans-fected with WT or ARE-deleted CCRK promoter constructs and Renilla luciferase reporters. After 24 h, the cells were further treated with human recombinant IL-6 for 3 h. For CCRK self-reinforced transcription, cells were co-transfected with CCRK construct, WT or ARE-deleted CCRK promoter constructs and Renilla luciferase reporters for 48 h. The treated cells were harvested using passive lysis buffer for 15 min and assayed by the Dual Luciferase Reporter Assay System (Promega) using GloMax microplate luminometer (Promega). The experiments were replicated three times in two independent experiments.

**Lipid accumulation assay.** A free fatty acid (FFA) mixture was prepared at 2:1 ratio of oleic acid (OA) to palmitic acid (PA) that mimics benign chronic steatosis with low toxicity described previously[38]. Briefly, 100 mM PA (Sigma) and 100 mM OA (Sigma) stocks were prepared in absolute ethanol at 70 °C and filter-sterilized. Five percent (w/v) FFA-free BSA solution was prepared in double-distilled water and filter-sterilized. A 5 mM stock solution for each fatty acid was prepared in 5% BSA solution in distilled water at 37 °C then the mixture was stored at 4 °C for further experiments. Nile Red (Sigma) is a dark purplish-red powder, the stock solution was prepared in DMSO at 1 mg/ml and then diluted 1:500–1:5000 in PBS for immunofluorescence staining of lipids. Cells were plated on coverslips in six-well plates and grown overnight. On the following day, cells were treated by conditional medium (DMEM supplemented with 10% FBS containing doxycycline or control solution with or without 10 nM rapamycin) for 36 h and starved for 12 h with FBS-free conditional medium. The starved cells were co-cultured with 200 μM FFA for 24 h to induce steatosis, and the BSA solution was used as control. Cells grown on coverslips were washed with PBS twice and fixed with 1 ml (for each well) of 3.7% formaldehyde/PBS for 15 min at room temperature. After this, cells were washed with PBS twice and stained with Nile-red solution for 30 min (samples were covered with foil to avoid bleaching by room light). Nuclei were counterstained by Hoechst (H33342, Calbiochem). The positive cells were calculated at random at ×400 under the fluorescence microscope and more than 1000 cells were assessed. The ratio of positive cells in every groups was used for statistical analysis. The experiments were replicated three times in three independent experiments.

**Insulin resistance assay.** Cells were plated in six-well plates and grown overnight. On the following day, cells were treated by conditional medium (DMEM supple-mented with 10% FBS-containing doxycycline or control solution with or without 10 M rapamycin) for 24 h and starved for 24 h with FBS-free conditional medium. Then, the starved cells were co-cultured with or without 100 nM insulin (Sigma) for 24 h to induce insulin resistance. In the last 1 h, the cells were incubated with fresh DMEM and treated briefly for 10 min with or without 10 nM insulin. The cells were then harvested, and the lysates were prepared as described in western blot assay (10 μl lysates blotted with indicated antibodies).

**Generation of conditional activation of CCRK TG mouse.** To overexpress CCRK specifically in adult mouse liver, a liver-specific transferrin promoter (pTf) together with transcription stopping cassette flanked by loxP sites (LSL) were used to control the expression of human CCRK. The 0.6 kb mouse transferrin promoter from C57BL/6 J genome DNA was amplified and subcloned in front of the 1.5 kb human CCRK cDNA in pcDNA3.1. The LSL stopping cassette was amplified from the genomic DNA of Rosa26LacZ mouse, and was inserted between the 0.63 kb transferrin promoter and the human CCRK cDNA. The linearized DNA carrying pTF-LSL-CCRK was released by PmeI, diluted into 1 ng/μl by TE buffer (10 mM Tris pH 7.5, 0.1 mM EDTA), before being subjected to Pronuclear injection. The expression of CCRK in Tg(pTf-LSL-CCRK) was blocked by the transcription stopping cassette. Removal of the stop cassette by tamoxifen-induced Cre recom-binase will turn on the expression of CCRK in a time-specific manner in mouse liver. To achieve this, Rosa26-CreERT2 mice were used to cross with Tg(pTf-LSL-CCRK) to establish the mouse line Rosa26-CreERT2/+; Tg(pTf-LSL-CCRK)/+. Primary hepatocytes were isolated from control and CCRK TG mouse as previous described. Adenovirus coding with Cre was used to infect primary hepatocytes for 4 h and then washout. Cells were collected 48, 96 and 144 h post infection for further analysis.

**NASH and NASH-HCC mouse model.** The obesity-promoted NASH and HCC models were constructed as described previously[7,30]. For the dietary obesity-NASH model, 6-week-old C57BL/6 male mice were randomly assigned to receive regular

CD or high fat high carbohydrate diet (HFHC; Surwit diet) and drinking water enriched with high-fructose corn syrup for 22 weeks. For the dietary obesity-HCC model, C57BL/6 male mice were injected with DEN (25 mg/kg; Sigma-Aldrich) at 14 days of age. After 4 weeks, mice were separated into two dietary groups and fed with CD or HFHC and drinking water enriched with high-fructose corn syrup for 22 weeks. Lentiviruses encoding shRNA against Ccrk (shCcrk) or control sequence (shCtrl) were packaged according to the manufacturer's instructions (Dharmacon) for transduction in the dietary obesity models. At the age of 6 and 18 weeks, $5 \times 10^7$ transducing units of lentiviruses in 100 ml PBS were administered via tail vein injection as previously described. Intraperitoneal glucose or insulin tolerance test (IPGTT/IPITT) was performed at the age of 26 weeks. All mice were sacrificed at the end of 28 weeks. All animal experimentation ethics approvals had been obtained from the Chinese University of Hong Kong (CUHK) Animal Experi-mentation Ethics Committee.

**Intraperitoneal glucose and insulin tolerance tests.** For intraperitoneal glucose tolerance test (IPGTT), 26-week-old mice were transferred to new clean cages and fasted overnight for ~16 h. On the following day, 1.5 g glucose per kg body weight was treated intraperitoneally. For intraperitoneal insulin tolerance test (IPITT), 26-week-old mice were transferred to new clean cages and fasted for 6 h. 75 U insulin per kg body weight was treated intraperitoneally. Tail blood was then collected under normal condition at indicated time-points. Glucose levels were measured using one touch blood glucose strips (Johnson and Johnson).

**Metabolic profiling.** Blood was collected from heart or tail vein, and kept at room temperature for not more than 4 h. The blood was centrifuged at 3000 rpm, 15 min, room temperature, and the serum was transferred to new tubes and stored at −20 ° C until use. The concentration of serum insulin, triglyceride, and non-essential fatty acid (NEFA) were measured using respective ELISA kits (LabAssay) according to the manufacturer's instructions.

**Xenograft mouse models.** For the xenograft mouse model, 6-week-old female athymic nude mice were injected with $5 \times 10^5$ LO2-CCRK-shRaptor and LO2-CCRK-shCtrl cells to the right flanks. 1 week after implantation, the drinking water of doxycycline-treated group mice was replaced by Dox drinking water containing 5% sucrose and 2 mg/ml doxycycline, which was changed every 3 days. The control group mice were treated by sucrose-enriched drinking water (5% sucrose) lacking doxycycline. For the rapamycin treatment, the freshly-made rapamycin (20 mg/ml stock in ethanol) diluted in 0.25% polyethylene glycol plus 0.25% Tween-20 was IP injected three times per week at 6 mg/kg, as previously described. Vehicle in the same volume was injected as control. Tumor length and width were measured using clipper and then tumor volume was calculated using the formula $V = (L \times W \times W)/2$. Tumor size was measured every other day using a caliper, and then tumor volume was calculated using the formula $V = (L \times W \times W)/2$, with $L$ indicating length and $W$ indicating width. The mice were sacrificed after 1 month for tumor size measurement.

**Ki67 immunohistochemistry.** Tissue sections (5 mm) from formalin-fixed par-affin-embedded liver tissues were prepared with a microtome (Leica). Then, the liver sections were deparaffinized, rehydrated, and washed in distilled water. Antigen retrieval was done using a pressure cooker with EDTA antigen retrieval buffer (1 mM EDTA, pH 8.0) for 10 min and rinsed by PBS. The endogenous peroxidase activity was then blocked by incubating the slides in 3% hydrogen peroxide in distilled water for 10 min at room temperature and rinsed using PBS. The sections were then blocked using blocking buffer for 30 min at room tem-perature and stained with polyclonal antibody against Ki-67 (1:100, Labvision) at 4 °C for 16 h. The universal HRP Multimer Ultraview Kit on Benchmark XL (Ventana Medical System) was used for chromogen development. For the nuclei staining, the sections were stained with the alum haematoxylin for 2 min, differentiated with 0.3% acid alcohol for several seconds and blue-up with Scott's tap water substitute for 1 min. The Ki67 indexes were assessed by counting the percentages of nucleus-stained (brown) cells/loci per total number of cells. At least 10 fields (magnification ×400) were counted per each liver section and the average indexes were calculated.

**CRISPR/Cas9-mediated genome editing.** Ccrk-depleted Hepa1–6 cells were generated by transfection with specific sgRNAs to Ccrk (sgRNA1, CGCCCTCCCCGATGCGACCG; sgRNA2, TGGCTTTGAAGACGATGCCG; sgRNA3, AGCACAAAGCCCGCACCATG) and selected by G418. Single clone was sorted by FACSArial Fusion (BD Biosciences) as previously described[27]. Ccrk depletion was confirmed by sequencing and Western blot analysis.

**Orthotopic HCC mouse model and MDSC expansion analysis.** Six-week-old CCRK transgenic mice were injected with tamoxifen (100 mg/kg body weight) daily for 3 consecutive days. At 30-day post tamoxifen injection, $5 \times 10^6$ Ccrk-depleted Hepa1–6 cells were intrahepatically injected into the liver of male CCRK TG mice. Mice were sacrificed at one month post-tumor injection. Blood, tumor, and matched non-tumor liver tissues were harvested. Single cell was isolated from

tumor or matched non-tumor tissues by gentleMACSTM dissociator (Miltenyi Biotec). Cells were stained with anti-mouse myeloid markers CD11b, Gr-1, Ly6C, Ly6G (eBioscience and BD Biosciences) and analyzed by flow cytometry using FACSFusion (BD Biosciences) and FlowJo software (Tree Star).

**T cell proliferation assay**. For autologous T cell proliferation assay, CD3$^+$CD8$^+$T-cells were labeled with carboxyfluorescein succinimidyl ester (CFSE; 5 µmol/L; Invitrogen) and co-cultured with CD11b$^+$Gr-1$^+$Ly6G$^+$Ly6C$^{Int}$ PMN-MDSCs (1:1) from liver tissues in the presence of PMA (50 ng/ml), inomycin (500 ng/ml), and recombinant IL-2 (R&D) for 5 days[27]. T cells with or without stimulation was used as positive or negative control, respectively. Surface staining for CD3/CD4/CD8 T cell markers and CFSE signals on T cells were acquired by flow cytometry using FACS Fortessa/FACSAria Fusion (BD Biosciences). The percentages of proliferating cells were determined and calculated by FlowJo software (Tree Star).

***Raptor* knockdown in *CCRK* TG mouse model**. Six-week-old male *CCRK* TG mice were injected with tamoxifen (100 mg/kg body weight) daily for 3 consecutive days. At 2-week post tamoxifen injection, $5 \times 10^7$ transducing units of shRaptor lentivirus were injected to *CCRK* TG mice by tail vein injection. Two weeks later, $5 \times 10^6$ *Ccrk*-depleted Hepa1–6 cells were intrahepatically injected into the liver of mice. Second dose of lentivirus was given to mice 2 weeks after surgery. Mice were sacrificed at one-month post-tumor injection. Blood, tumor, and matched non-tumor liver tissues were harvested. Single cell was isolated from tumor or matched non-tumor tissues by gentleMACSTM dissociator (Miltenyi Biotec). Cells were stained with anti-mouse myeloid markers CD11b, Gr-1, Ly6C, Ly6G (eBioscience and BD Biosciences) and analyzed by flow cytometry using FACSFusion (BD Biosciences) and FlowJo software (Tree Star).

**MDSC depletion in *CCRK* TG mouse model**. Six-week-old male *CCRK* TG mice were injected with tamoxifen (100 mg/kg body weight) daily for 3 consecutive days. At 30-day post tamoxifen injection, $5 \times 10^6$ *Ccrk* KO Hepa1–6 cells were intrahepatically injected into the liver of mice. For MDSC depletion, three doses of anti-Ly6G antibody (Bio X Cell) were intraperitoneally injected to mice every 5 days[27]. Mice were sacrificed at one-month post-tumor injection. Blood, tumor, and matched non-tumor liver tissues were harvested. Single cell was isolated from tumor or matched non-tumor tissues by gentleMACSTM dissociator (Miltenyi Biotec). Cells were stained with anti-mouse myeloid markers CD11b, Gr-1, Ly6C, Ly6G (eBioscience and BD Biosciences) and analyzed by flow cytometry using FACS-Fusion (BD Biosciences) and FlowJo software (Tree Star).

**Nanoparticle-mediated hepatic IL-6 depletion**. A lipid/calcium/phosphate (LCP) nanoparticle optimized for delivering plasmid DNA to the nucleus of liver hepatocytes was delivered via asialoglycoprotein receptor (ASGPR). The coding sequence of IL-6 receptor-binding domain was used to generate plasmid DNA and subsequently encapsulated into LCP nanoparticle. The LCP loaded with plasmid DNA of control GFP (pGFP) or IL-6 (pIL-6-trap) was prepared using a modified protocol as described previously[63]. The formulation of pGFP and pIL-6-trap, including a His(6×)-Tag at the C-terminal end, was injected (0.25 ml, balanced in osmolarity with the addition of sucrose) into the orthotopic liver tumor-bearing C57Bl/6 mice at a dose of 30 µg/each through tail vein at a 5-day interval[27]. Mice were then sacrificed at day-28 post-tumor implantation. The IL-6 concentration in liver tissues was detected by IL-6 high sensitive ELISA kit (R&D). Liver and tumor tissues were collected for further analysis.

**Statistical analysis**. Unless otherwise indicated, data are presented as mean ± SD from three independent experiments. The GraphPad Prism 5 (GraphPad Software, San Diego, CA) was used for statistical analysis. The independent Student's $t$-test or non-parametric test was used to compare gene expression and functional parameters between two selected groups. ANOVA followed by Bonferroni post hoc tests were performed when more than two groups were included in the experimental results. The transcript levels of *IL-6*, *CCRK*, *CD33*, *OLR1* and the protein levels of CCRK, p-GSK3β$^{Ser9}$/GSK3β, p-mTOR$^{Ser2448}$/mTOR, p-4E-BP1$^{Thr37/46}$/4E-BP1, and p-S6K$^{Thr389}$/S6K, mSREBP1, p-STAT3$^{Tyr705}$/STAT3, and AR in the paired tumor and non-tumor tissues were compared using nonparametric Wilcoxon matched pairs test. The correlation between mRNA or protein expression was assessed using Pearson's Chi-square test. A two-tailed $p$-value of <0.05 was regarded as statistically significant.

## Data availability
All data supporting the findings of this study are available within the article and Supplementary Information, or from the corresponding author upon request.

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

## Acknowledgements

This project is supported by the University Grants Committee through the Collaborative Research Fund C4017-14G, General Research Fund 14120816 and 14102914, the Food and Health Bureau through Heath and Medical Research Fund 03141376, the Focused Innovations Scheme B 1907309 from the Chinese University of Hong Kong (CUHK), and the Li Ka Shing Foundation (Canada). A.S.L.C. is supported by funding from the Young Researcher Award, CUHK.

## Author contributions

Study concept and design: G.L.H.W., V.W.S.W., and A.S.L.C.; acquisition of data: H.S., W.Y., Y.T., X.Z., J.Z., M.T.S.M., W.T., Y.F., L.X., A.W.H.C., and B.Y.; analysis and interpretation of data: H.S., W.Y., Y.T., X.Z., J.Z., M.T.S.M., W.T., Y.F., L.X., A.W.H.C., and A.S.L.C.; acquisition of patient specimens: A.W.H.C., J.H.T., Y.S.C., and P.B.S.L.; drafting of the manuscript and preparation of figures: H.S., W.Y., X.Z., M.T.S.M., and A.S.L.C.; critical revision of the manuscript: K.F.T., J.J.Y.S., G.L.H.W., V.W.S.W., and A.S.L.C.; obtained funding: K.L.C., K.F.T., J.J.Y.S., G.L.H.W., V.W.S.W., and A.S.L.C.; administrative, technical or other material support: H.K.S.W., S.W.T., K.L.C., M.H., R.L., L.H., and P.Y.; study supervision: G.L.H.W., V.W.S.W., and A.S.L.C.

## Additional information

**Competing interests:** G.L.H.W. has served as an advisory committee member for Gilead Sciences, and as a speaker for Abbott, Abbvie, Bristol-Myers Squibb, Echosens, Furui, Gilead Sciences, Janssen and Roche. V.W.S.W. has served as an advisory board member for AbbVie, Allergan, Center for Outcomes Research in Liver Diseases, Gilead Sciences, Janssen, Perspectum Diagnostics and Pfizer; and received lecture fees from Bristol-Myers Squibb, Echosens, Gilead Sciences, and Merck. The remaining authors declare no competing interests.

