## [Peer Review File · Nature Communications]

Reviewers' Comments:

Reviewer #1:

Remarks to the Author:

This manuscript describes the functional contribution of CCRK in NASH-associated HCC pathogenesis. Specifically, authors demonstrated that CCRK is upregulated in obese mouse liver tissues by IL6/Stat3 and AR cascades. Knockdown of CCRK inhibited obesity associated HCC development in mice. Mechanistically, it was found that CCRK activates TSC2/mTORC1 via p-GSK3; CCRK promotes HCC formation requires mTORC2. Furthermore, CCRK-mTORC1 recruits MDSCs in the pro-tumoral microenvironments. Finally, human studies demonstrate the correlation between CCRK overexpression and mTORC1 activation in NASH-HCCs. Overall this is a solid study and it provides novel insight into how CCRK regulates NASH-related hepatocarcinogenesis. Some issues, however, need to be addressed.

1. Overexpression of CCRK was done L02 and SK-Hep1 cells. HCC research community is no longer accepting SK-Hep1 cells as it is NOT a liver cancer cell lines (Rebouissou S, et al, JHep 2017; [http://www.journal-of-hepatology.eu/article/S0168-8278\(17\)32206-7/fulltext](http://www.journal-of-hepatology.eu/article/S0168-8278(17)32206-7/fulltext)). Those experiments should be performed using a different HCC cell line.
2. It is very impressive the group is able to silence a gene in the mouse liver via lentiviral shRNA delivery. In most reports, it seems that a PHx is required for high efficiency lentivirus based gene delivery in the liver. It would be helpful to inhibit mTORC1 in the mouse liver via lentiviral shRaptor in CCRK Tg mice to determine whether indeed the recruitment of MDSCs and enhanced tumorigenic potential of Hepa1-6 cells require mTORC1.
3. The role of MDSC cells in mediating CCRK induced HCC is not entirely convincing since the data are rather correlated. It would ideal if the authors could deplete MDSC cells with the antibody and test whether CCRK Tg mice are still able to promote HCC development.
4. CD33 is not a specific human marker for MDSC cells, and human PMN-MDSC cells have CD33dim staining (Dumitru CA, et al, 2012). The result in human NASH related HCC is not conclusive.
5. If CCRK is able to induce mTORC1/SREBP cascade, does CCRK Tg mice show fatty liver phenotype (as those seen in Pten KO mice)?
6. Since GSK3 β regulates Wnt/ β -catenin signaling, what is the Wnt/ β -catenin signaling status in NASH related HCC? And whether it is regulated when CCRK is knocked down?

Reviewer #2:

Remarks to the Author:

In this interesting manuscript the authors are trying to identify the molecular mechanism why men are more likely to develop HCC in the context of obesity and NASH. They identify CCRK as the central mediator responsible for lipid accumulation and other metabolic events, HCC development and changes in the tumor microenvironment leading to enhanced immunosuppression. Results are supported by data from human specimen.

1. The authors miss an important control throughout the manuscript: Tumor studies should be tested in Ccrk knockdown mice without NASH. It is not clear from the study what the specific effect Ccrk has on NASH vs HCC development.
2. The authors claim that the results provided explain why males are more likely to develop HCC in the context of NASH (AR dependent mechanism): If this is really the case, please provide data showing that one cannot find enhanced CCRK activation in human NASH samples from female patients in contrast to male patients and conduct both in vitro studies using cell lines derived from female mice and in vivo studies in female mice on HFHC diet.
3. Authors claim that they find an accumulation of MDSC based on phenotypic analysis. However functional data is needed to demonstrate immunosuppressive function.
4. Again the question comes up about MDSC in female mice on HFHC diet.
5. Human samples are poorly characterized. Please describe extent of NASH using the scoring system described by Kleiner et al.

Reviewer #3:

Remarks to the Author:

This manuscript investigated the roles CCRK in NASH and NASH-associated HCC development. The authors showed that IL-6 upregulation in NASH environment triggered a feedforward loop of STAT3-AR-CCRK, which activated mTORC1-SREBP lipogenic pathway through GSK3b phosphorylation and resulted in aggravation of hepatic steatosis and HCC development. The authors also showed that CCRK induced tumor immune evasion by recruitment of MDSC through mTORC1 activation.

Although these results have potential interest and the experiments were well designed and performed, many data presented herein had already been published. In addition, the authors don't refer appropriately to already published findings as mentioned below. The novel finding in this paper is the role of CCRK in NASH progression, and in my opinion, it's better to focus on this point.

1. The authors showed that lentiviral-mediated knockdown of CCRK significantly suppressed DEN-induced HCC development under HFHC diet. However, knockdown of CCRK using the same method was previously reported to inhibit markedly DEN-induced HCC development under normal diet (Feng, *J Hepatology* 2015). Therefore, to conclude that CCRK inhibition suppresses obesity-mediated promotion of DEN-induced hepatocarcinogenesis, comparison of HFHC diet model with normal diet model is mandatory. Although the authors cited the paper by Feng et al., the result of DEN model was not mentioned at all in the manuscript.

2. The authors showed a feedforward loop of STAT3-AR-CCRK in NASH, which could activate mTORC1-SREBP lipogenic pathway through GSK3b phosphorylation. A previous study reported by Yu et al, showed that CCRK-mediated GSK3b phosphorylation induced b-catenin-TCF-AR regulatory loop which played a critical role in HBV-associated hepatocarcinogenesis (*Gut* 2014). Although the authors cited this paper, the b-catenin-TCF-AR regulatory loop was not mentioned at all in the manuscript. The authors should analyze the implication of b-catenin-TCF-AR regulatory loop in obesity-associated hepatocarcinogenesis and clarify whether the downstream signaling of CCRK differs according to the underlying etiology.

3. The authors showed that the enhanced CCRK expression in HCC induced the tumor immune evasion by recruitment of MDSC through mTORC1 activation. On the other hand, although Zhou, et al. also reported that CCRK in HCC inhibited anti-tumor T cell response by recruitment of MDSC, this effect was exerted through NFkB activation (*Gut* 2017). Although this paper was cited in the manuscript, implication of NFkB activation was not mentioned. The authors should refer appropriately to already published findings and analyze implication of NFkB in MDSC recruitment.

4. The authors showed increased expression of CCRK in HCC, but the increased CCRK in HCC was already reported in several studies (Feng, *JCI* 2011; Yu, *Gut* 2014; Feng, *J Hepatology* 2015). The expression of CCRK was assessed in only NASH-associated HCC in this manuscript, are there any differences in the expression levels of CCRK among underlying etiologies? In addition, according to the results of mouse experiments, CCRK expression may be increased in NASH or NAFLD especially in male. Furthermore, the relationship between CCRK and NASH severity and also a gender difference of CCRK expression in NASH patients are potentially interesting.

5. In Figure 1G and H, CCRK knockdown markedly suppressed HFHC diet-induced liver steatosis, and that was explained by inhibition of mTORC1-SREBP activation and subsequent lipogenesis. However, because this diet contains a high amount of fat, inhibition of de novo lipid synthesis may not be sufficient to explain such a marked reduction of liver steatosis. To examine the implication of de novo fatty acid synthesis in HFHC-induced liver steatosis, the expression of SREBP target genes in the liver from HFHC diet-fed mice with control or CCRK shRNA should be analyzed.

Additionally, the involvement of CCRK in fatty acid uptake, fatty acid beta-oxidation, and lipid secretion from hepatocytes also should be explored.

7. The authors showed the activation of mTOR-SREBP pathway in the liver of CCRK Tg mice. If so, do CCRK Tg mice develop spontaneous liver steatosis?

8. Although the authors showed that IL-6 increased the expression of CCRK via STAT3 activation in vitro, in vivo evidence supporting the key role of IL-6 in triggering CCRK-mediated feedforward loop was missing, such as IL-6 KO or IL-6 inhibitor data.

Minor comments

1. The pictures of liver histology were too small. Especially, it was very difficult to identify spotty necrosis in Figure 1G.

2. In DEN model, the data of CCRK expression in the tumor tissues was not found. Comparison of CCRK expression in HCC tissues between normal diet and HFHC diet may also be informative.

3. In Figure 3K, the quality of western blotting of AR protein in L-02 cells was not enough for publication.

Reviewer #4:

Remarks to the Author:

The main theme in this manuscript is its attempt to explain hepatocellular carcinoma (HCC) mechanistically, by drawing arrows among a multitude of signaling molecules, linking inflammation to constitutive mTORC1 signaling, which is known to be involved in promoting non-alcoholic steatohepatitis (NASH) and HCC.

The novelty of the study is placing cell cycle-related kinase (CCRK) at the center of this mechanism, since its regulation by the Androgen Receptor (AR) could explain the predominance of obesity-induced HCC in males compared to females.

I find the described mechanism a very complicated one: it details a complex, intertwined signal regulation involving IL6-induced expression of CCRK through phosphorylation of STAT3 and upregulation of the Androgen Receptor (AR). CCRK can then itself induce a feed-forward mechanism that enhances the binding of STAT3-Androgen receptor (AR) to its promoter, further increasing its expression.

Increased CCRK levels cause increased phosphorylation of GSK3beta on Ser9, inhibiting its activity towards TSC2, which in turn, results in mTORC1 activation and increased SREBP-1 activity, leading to increased lipid synthesis in the liver, thus promoting non-alcoholic fatty liver disease (NAFLD), and HCC.

Aside from CCRK, the novelty of these findings is mild to moderate, as the story proposes a hypothesis that could piece the puzzle together, using as a basis, what is already known in the literature to be involved in NASH and HCC.

Some minor, yet relevant comments concerning the data presented:

1- The "n" number of samples should be listed per group in each experiment.

2- Student's t-test was employed to compare 2 groups even when more than 2 groups were included in the experimental results; this is not statistically appropriate. Even when "comparing" 2 groups, if more than 2 are within the same experimental set-up (or graphs), ANOVA should be performed, followed by a post-hoc test, such as Newman-Keuls or Bonferroni, which would then compare any 2 groups of interest.

This is applicable to all figures. For example: 1C-F, 1H-J, 2C, 2E-K etc...

3- When assessing glucose tolerance or insulin tolerance by GTT or ITT, quantitative graphs should represent the "area under the curve", Not a picked time-point of interest. The authors, for instance, chose 15min timepoint (Fig. 1D, E and 2K). It is obvious that there are very mild (if any) changes in glucose tolerance or insulin tolerance in Figures 1E and 2K.

4- A bit more detail is required to describe the Figure legends, in general especially when not clarified in the text: for example, describe the samples in Fig. 3L, 4C, 4D and 7B.

Response to Reviewer #1:

We appreciate the reviewer's compliment on our '*solid study*' which '*provides novel insight into how CCRK regulates NASH-related hepatocarcinogenesis*'.

Major comments:

1. Overexpression of CCRK was done LO2 and SK-Hep1 cells. HCC research community is no longer accepting SK-Hep1 cells as it is NOT a liver cancer cell lines (Rebouissou S, et al, JHep 2017; [http://www.journal-of-hepatology.eu/article/S0168-8278\(17\)32206-7/fulltext](http://www.journal-of-hepatology.eu/article/S0168-8278(17)32206-7/fulltext)). Those experiments should be performed using a different HCC cell line.

Response:

We thank the reviewer for pointing out the SK-Hep1 cell identity issue. As mentioned in the captioned editorial¹ and a recent report², SK-Hep1 is derived from liver sinusoidal endothelial cells (LSECs) instead of liver cancer cells. In our study, SK-Hep1 cell line was mainly used to determine the molecular function of CCRK in ectopic expression experiments due to its low endogenous CCRK expression. As all the HCC cell lines tested are expressing relatively high level of CCRK, we have performed additional experiments using a Huh7 HCC cell model with CRISPR/Cas9-mediated CCRK knockout (KO). Our data from the CCRK KO Huh7 HCC cells showed that ectopic expression of CCRK, but not its kinase-defective mutant, up-regulated STAT3 phosphorylation and AR expression (new **Figure 3F**). Moreover, overexpressed CCRK phospho-inactivated GSK3 β to up-regulate mTORC1/4E-BP1/S6K signaling, which could be abolished by co-expression of the constitutively active S9A-GSK3 β mutant (new **Figure 4A**). These results corroborate with our previous data using a LO2 immortalized hepatocyte line to validate the inflammatory-CCRK circuitry and its key molecular function in mTORC1 activation. As SK-Hep1 can serve as a cell model for LSECs, which play key roles in the development of chronic liver diseases and HCC³, we have moved the SK-Hep1 data to Supplementary Figure 3 with specification of its cell identify (page 8).

2. It is very impressive the group is able to silence a gene in the mouse liver via lentiviral shRNA delivery. In most reports, it seems that a PHx is required for high efficiency lentivirus based gene delivery in the liver. It would be helpful to inhibit mTORC1 in the mouse liver via lentiviral shRaptor in CCRK Tg mice to determine whether indeed the recruitment of MDSCs and enhanced tumorigenic potential of Hepa1-6 cells require mTORC1.

Response:

We thank the reviewer for the compliment and comment. To investigate whether the mTORC1 signaling promotes MDSC recruitment and tumorigenicity in CCRK TG mice, we applied the lentiviral shRNA-mediated Raptor (shRaptor) knockdown approach. We found that down-regulation of Raptor in the livers of TG mice (new **Supplementary Figure 5C**)

abolished tumor growth ($p < 0.05$; new **Figure 6D**) and significantly reduced PMN-MDSCs ($p < 0.05$; new **Figure 6E**). Moreover, the level of PMN-MDSCs, but not M-MDSCs, in the liver was positively correlated with tumor weight in the *CCRK* TG mouse model ($p < 0.005$; new **Figure 6F** and **Supplementary Figure 5B**).

3. *The role of MDSC cells in mediating CCRK induced HCC is not entirely convincing since the data are rather correlated. It would ideal if the authors could deplete MDSC cells with the antibody and test whether CCRK Tg mice are still able to promote HCC development.*

Response:

To address this important comment, we have depleted MDSCs by intraperitoneal injection of anti-Ly6G antibody⁴ after tumor cell implantation in *CCRK* TG mice (new **Supplementary Figure 5D**). Administration of anti-Ly6G antibody significantly suppressed the accumulation of PMN-MDSCs in liver ($p < 0.01$), leading to a borderline significant reduction in HCC tumorigenicity ($p = 0.0555$; new **Supplementary Figure 5E-F**). This new data supports the functional significance of MDSCs in mediating *CCRK*-induced HCC.

4. *CD33 is not a specific human marker for MDSC cells, and human PMN-MDSC cells have CD33dim staining (Dumitru CA, et al, 2012). The result in human NASH related HCC is not conclusive.*

Response:

We have performed quantitative RT-PCR on a specific marker of human PMN-MDSCs i.e. lectin-type oxidized LDL receptor encoded by *OLRI*⁵ using the same cohort of patient samples with available cDNA. Consistent with *CD33*, we found that *OLRI* expression was concurrently up-regulated with *CCRK* in human NASH-related HCCs (new **Supplementary Figure 6A-B**).

5. *If CCRK is able to induce mTORC1/SREBP cascade, does CCRK Tg mice show fatty liver phenotype (as those seen in Pten KO mice)?*

Response:

The fatty liver phenotype of *CCRK* TG and control mice fed with chow or HFHC diet was examined using H&E stained liver sections and graded by the NASH Clinical Research Network scoring system⁶. We found that the HFHC diet-fed *CCRK* TG mice exhibited more extensive steatosis than the control mice ($p < 0.05$; new **Supplementary Figure 4A-B**). This result indicates that mice with hepatic *CCRK* over-expression are more prone to fatty liver development, which is consistent with the *in vitro* finding that *CCRK*-expressing LO2 hepatocytes with fatty acid exposure exhibited marked increase in lipid accumulation as compared to non-expressing cells ($p < 0.001$; Figure 5B-C).

6. Since GSK3 β regulates Wnt/ β -catenin signaling, what is the Wnt/ β -catenin signaling status in NASH related HCC? And whether it is regulated when CCRK is knocked down?

Response:

To investigate whether Wnt/ β -catenin signaling is perturbed in NASH-related HCC, we examined the levels of active and total β -catenin in the dietary obesity HCC model by Western blot. In accordance with CCRK up-regulation and GSK3 β phosphorylation, the active β -catenin was concordantly increased in the liver tissues of NASH-related HCC (new **Supplementary Figure 7**). Moreover, down-regulation of *Ccrk* markedly suppressed the signaling (new **Supplementary Figure 7**), thus implying an involvement of CCRK/GSK3 β / β -catenin signaling in NASH-related hepatocarcinogenesis.

Response to Reviewer #2:

We appreciate the reviewer's compliment on our '*interesting manuscript*' in which '*results are supported by data from human specimen*'.

Major comments

1. The authors miss an important control throughout the manuscript: Tumor studies should be tested in *Ccrk* knockdown mice without NASH. It is not clear from the study what the specific effect *Ccrk* has on NASH vs HCC development.

Response:

To address this important comment, we have now included the tumor data of *Ccrk* knockdown mice without NASH i.e. administration of low-dose diethylnitrosamine (DEN) at 14 days of age and fed with chow diet. This experiment was performed side-by-side with the NASH-related HCC model induced by DEN plus HFHC diet, and all mice were sacrificed at the end of 28 weeks. We found that *Ccrk* knockdown reduced HCC multiplicity and tumor size (by ~50%) in the DEN model (new **Supplementary Figure 2A-B**), though the differences did not reach statistical significance in contrast to the DEN plus HFHC diet model. In a previous study in which the DEN model lasted for 32 weeks, which resulted in significantly more (~4-fold) and larger (~5-fold) tumors than that for 28 weeks in the present study, *Ccrk* knockdown by the same lentiviral approach significantly reduced HCC tumorigenicity⁷. Therefore, CCRK can promote hepatocarcinogenesis in both NASH and non-fatty liver contexts. Our new data demonstrated that CCRK promotes HCC development in part through the GSK3 β - β -catenin-AR regulatory loop in both the HFHC diet (new **Supplementary Figure 7**) and chow diet models as shown previously⁷. Moreover, we showed that CCRK contributes to NASH and NASH-related hepatocarcinogenesis through mTORC1-mediated lipid metabolic (Figure 5A-G and new **Supplementary Figure 2E-H**) and immune dysregulation (new **Figure 6C-F**).

2. *The authors claim that the results provided explain why males are more likely to develop HCC in the context of NASH (AR dependent mechanism): If this is really the case, please provide data showing that one cannot find enhanced CCRK activation in human NASH samples from female patients in contrast to male patients and conduct both in vitro studies using cell lines derived from female mice and in vivo studies in female mice on HFHC diet.*

Response:

We have obtained additional liver biopsy tissues from 13 male and 10 female NAFLD patients (new **Supplementary Table 2**) for qRT-PCR analysis. Although no paired normal liver tissues could be obtained due to ethical reason, we found that the *CCRK* transcript level in male steatosis/NASH tissues was significantly higher than the female tissues ($p < 0.05$; new **Supplementary Figure 6C**). Since the murine Hepa1-6 cells used in this study were derived from hepatoma developed in a male C57/L mouse⁸, we have performed *AR* ectopic expression or knockdown in different cell models to address the gender effect, showing that IL-6/STAT3 signaling activates *CCRK* in an *AR*-dependent manner (Figure 3). Moreover, we have shown that HFHC-fed female mice did not exhibit *CCRK* induction, triglyceride/NEFA abnormalities, and glucose intolerance (Supplementary Figure 1) in contrast to the HFHC-fed male mice.

3. *Authors claim that they find an accumulation of MDSC based on phenotypic analysis. However functional data is needed to demonstrate immunosuppressive function.*

Response:

To demonstrate the immunosuppressive function of MDSC in our model, PMN-MDSCs sorted from the liver of *CCRK* TG and control mice were co-cultured with allogeneic splenic CD3⁺ T cells. We found that the liver-infiltrating PMN-MDSCs from the TG mice exerted a strong inhibition of T cell proliferation compared to those of the control mice (new **Supplementary Figure 5A**).

4. *Again the question comes up about MDSC in female mice on HFHC diet.*

Response:

Interestingly, the level of PMN-MDSCs was significantly higher in male but lower in female *CCRK* TG compared to control mice on HFHC diet (new **Supplementary Figure 4C**). This finding concurs with the oncogenic role of *AR/CCRK* signaling via MDSC-mediated immunosuppression⁹.

5. *Human samples are poorly characterized. Please describe extent of NASH using the scoring system described by Kleiner et al.*

Response:

A new **Supplementary Table 1** describing the clinicopathological features of HCC patients, including the NAFLD scoring by Kleiner *et al.*, is provided in the revised version.

Response to Reviewer #3:

We appreciate the reviewer's compliment on the results with '*potential interest*' and '*the experiments (that) were well designed and performed*'.

Major comments

1. *The authors showed that lentiviral-mediated knockdown of CCRK significantly suppressed DEN-induced HCC development under HFHC diet. However, knockdown of CCRK using the same method was previously reported to inhibit markedly DEN-induced HCC development under normal diet (Feng, J Hepatology 2015). Therefore, to conclude that CCRK inhibition suppresses obesity-mediated promotion of DEN-induced hepatocarcinogenesis, comparison of HFHC diet model with normal diet model is mandatory. Although the authors cited the paper by Feng et al., the result of DEN model was not mentioned at all in the manuscript.*

Response:

We thank the reviewer for the comment. We have compared the tumor data of CCRK inhibition in DEN-treated mice fed with HFHC or chow diet and sacrificed at the age of 28 weeks. We found that *Ccrk* knockdown reduced HCC multiplicity and tumor size (by ~50%) in the chow diet model (new **Supplementary Figure 2A-B**), though the differences did not reach statistical significance in contrast to the HFHC diet model. As mentioned by the reviewer, *Ccrk* knockdown by the same lentiviral approach significantly reduced HCC tumorigenicity in a previous study in which the DEN model lasted for 32 weeks⁷, resulting in significantly more (~4-fold) and larger (~5-fold) tumors than that for 28 weeks in the present study. Therefore, CCRK can promote hepatocarcinogenesis in both NASH and non-fatty liver contexts. We further mentioned in the revised manuscript that CCRK also controlled GSK3 β - β -catenin-AR signaling in DEN-induced HCC development under HFHC diet similar to the DEN model in the HBV context¹⁰ (Page 17-18 in the revised version).

2. *The authors showed a feedforward loop of STAT3-AR-CCRK in NASH, which could activate mTORC1-SREBP lipogenic pathway through GSK3b phosphorylation. A previous study reported by Yu et al, showed that CCRK-mediated GSK3b phosphorylation induced b-catenin-TCF-AR regulatory loop which played a critical role in HBV-associated hepatocarcinogenesis (Gut 2014). Although the authors cited this paper, the b-catenin-TCF-AR regulatory loop was not mentioned at all in the manuscript. The authors should analyze the implication of b-catenin-TCF-AR regulatory loop in obesity-associated hepatocarcinogenesis and clarify whether the downstream signaling of CCRK differs according to the underlying etiology.*

Response:

We have performed additional Western blot analysis to examine the status of β -catenin and AR signaling in our DEN-induced HCC model under HFHC diet. The results showed that the active β -catenin and AR (denoted by p-AR^{ser81}) as well as the total AR levels were concordantly increased in the liver tissues of obesity-related HCC, but markedly suppressed by down-regulation of *Ccrk* (new **Supplementary Figure 7**). This data suggests that CCRK activates the GSK3 β - β -catenin-AR regulatory loop in both HBV¹⁰ and obesity-associated hepatocarcinogenesis, which may contribute to the lipogenic tumor phenotype¹¹ in the latter case. Overall, our study highlights CCRK as a crucial mTORC1 regulator in the development of NASH and NASH-associated HCC (Page 17-18 in the revised version).

3. The authors showed that the enhanced CCRK expression in HCC induced the tumor immune evasion by recruitment of MDSC through mTORC1 activation. On the other hand, although Zhou, et al. also reported that CCRK in HCC inhibited anti-tumor T cell response by recruitment of MDSC, this effect was exerted through NF κ B activation (Gut 2017). Although this paper was cited in the manuscript, implication of NF κ B activation was not mentioned. The authors should refer appropriately to already published findings and analyze implication of NF κ B in MDSC recruitment.

Response:

We thank the reviewer for pointing out the importance of NF- κ B signaling in CCRK-mediated tumor immune evasion via MDSC. We have performed additional Western blot analysis to examine the status of NF- κ B signaling in our DEN-induced HCC model under HFHC diet. We found that the expression of active p65 (denoted by p-p65^{ser536}, a major subunit of NF- κ B) was significantly increased in the liver tissues of obesity-related HCC, but abolished by knockdown of *Ccrk* (new **Supplementary Figure 7**). As the level of PMN-MDSCs was also normalized to basal level by *Ccrk* knockdown in the same model (Figure 6G-H), our data suggest that CCRK may simultaneously induce MDSC expansion and recruitment via NF- κ B/IL-6 and mTORC1/G-CSF signaling, respectively (Page 18 in the revised version).

4. The authors showed increased expression of CCRK in HCC, but the increased CCRK in HCC was already reported in several studies (Feng, JCI 2011; Yu, Gut 2014; Feng, J Hepatology 2015). The expression of CCRK was assessed in only NASH-associated HCC in this manuscript, are there any differences in the expression levels of CCRK among underlying etiologies? In addition, according to the results of mouse experiments, CCRK expression may be increased in NASH or NAFLD especially in male. Furthermore, the relationship between CCRK and NASH severity and also a gender difference of CCRK expression in NASH patients are potentially interesting.

Response:

To examine the potential difference of CCRK expression in HCC with different etiologies, we have compared the *CCRK* expression levels between human HBV- and NASH-associated

HCCs by qRT-PCR analysis. The results showed that *CCRK* was progressively increased from normal liver tissues to non-tumor liver tissues to HCCs in both HBV-infected and NASH patients (new **Supplementary Figure 6D**). Moreover, no significant difference between HBV- and NASH-associated HCCs was observed. Besides, we have obtained 9 steatosis and 14 NASH tissues from 13 male and 10 female patients (new **Supplementary Table 2**) for qRT-PCR analysis. We found that *CCRK* expression was significantly higher in male compared to female NAFLD tissues (new **Supplementary Figure 6C**). However, we did not observe significant difference between steatosis and NASH tissues in this small cohort (data not shown).

5. In Figure 1G and H, CCRK knockdown markedly suppressed HFHC diet-induced liver steatosis, and that was explained by inhibition of mTORC1-SREBP activation and subsequent lipogenesis. However, because this diet contains a high amount of fat, inhibition of de novo lipid synthesis may not be sufficient to explain such a marked reduction of liver steatosis. To examine the implication of de novo fatty acid synthesis in HFHC-induced liver steatosis, the expression of SREBP target genes in the liver from HFHC diet-fed mice with control or CCRK shRNA should be analyzed. Additionally, the involvement of CCRK in fatty acid uptake, fatty acid beta-oxidation, and lipid secretion from hepatocytes also should be explored.

Response:

We have examined the mouse liver tissues for the effects of *CCRK* on multiple lipid metabolic pathways by qRT-PCR analysis. We found that *Ccrk* down-regulation resulted in significant reduction of *Srebf1* and its target genes *acetyl-CoA carboxylase 1 (Acc1)*, *acetone-cyanohydrin lyase (Acl)* and *fatty acid synthase (Fasn)*, which are major drivers in *de novo* lipogenesis (new **Supplementary Figure 2E**). In parallel, *fatty acid binding protein 3 (Fabp3)* was down-regulated suggestive of reduced fatty acid uptake (new **Supplementary Figure 2F**), while *ATP binding cassette subfamily G member 1 (Abcg1)* was up-regulated indicative of enhanced lipid secretion (new **Supplementary Figure 2G**). Intriguingly, *carnitine palmitoyltransferase 2 (Cpt2)* and *peroxisome proliferator activated receptor alpha (Ppara)* were also down-regulated, denoting a reduction of fatty acid beta-oxidation (new **Supplementary Figure 2H**). Collectively, these data suggest that *CCRK*-induced liver steatosis in obesity-associated hepatocarcinogenesis is a net result of increased lipid synthesis and uptake, and decreased lipid secretion, which may compensate for the elevated fatty acid beta-oxidation.

6. The authors showed the activation of mTOR-SREBP pathway in the liver of CCRK Tg mice. If so, do CCRK Tg mice develop spontaneous liver steatosis?

Response:

The fatty liver phenotype of *CCRK* TG and control mice fed with chow or HFHC diet was examined using H&E stained liver sections and graded by the NASH Clinical Research Network scoring system⁶. We found that the HFHC diet-fed *CCRK* TG mice exhibited more extensive steatosis than the control mice ($p < 0.05$; new **Supplementary Figure 4A-B**). This

result indicates that mice with hepatic CCRK over-expression are more prone to fatty liver development, which is consistent with the *in vitro* finding that CCRK-expressing LO2 hepatocytes with fatty acid exposure exhibited marked increase in lipid accumulation as compared to non-expressing cells ($p < 0.001$; Figure 5B-C).

7. *Although the authors showed that IL-6 increased the expression of CCRK via STAT3 activation in vitro, in vivo evidence supporting the key role of IL-6 in triggering CCRK-mediated feedforward loop was missing, such as IL-6 KO or IL-6 inhibitor data.*

Response:

We have examined the IL-6/STAT3/CCRK signaling in hepatoma-bearing C57Bl/6 mice injected with or without IL-6-trap, a liver-specific nanoparticle carrying a gene for the expression of an IL-6-specific neutralization protein⁹. The liver tissues from these mice were then subjected to ELISA and Western blot analyses. The results showed that IL-6-trap significantly reduced IL-6 level in the mouse liver tissues (new **Supplementary Figure 3C**). Moreover, the p-STAT3^{Tyr705} and CCRK levels were dramatically down-regulated following IL-6 neutralization (new **Supplementary Figure 3D**). These *in vivo* data confirm our previous *in vitro* results that IL-6 increases CCRK expression via STAT3 activation (Figure 3).

Minor comments

1. *The pictures of liver histology were too small. Especially, it was very difficult to identify spotty necrosis in Figure 1G.*

Response:

We have now enlarged Figure 1G in the revised version.

2. *In DEN model, the data of CCRK expression in the tumor tissues was not found. Comparison of CCRK expression in HCC tissues between normal diet and HFHC diet may also be informative.*

Response:

We have performed Western blot using the tumor tissues of our DEN models. We found high tumoral CCRK expression but no obvious difference regardless of gene knockdown or diet (normal/HFHC) (data not shown). These results are consistent with the experimental scheme of lentiviral-mediated *Ccrk* knockdown during cancer initiation but not progression. The data also imply the importance of CCRK in establishing a pro-tumorigenic microenvironment for HCC development, as the dramatic reduction in tumorigenicity was associated with CCRK suppression in the peri-tumoral liver tissues (Figure 2C-D).

3. In Figure 3K, the quality of western blotting of AR protein in L-02 cells was not enough for publication.

We have repeated the experiment and improved the quality of the Western blot as shown in the new **Figure 3H** of the revised version.

Response to Reviewer #4:

We appreciate the reviewer's acknowledgement that '*the novelty of the study is placing cell cycle-related kinase (CCRK) at the center of...a multitude of signaling molecules, linking inflammation to constitutive mTORC1 signaling*', which '*could explain the predominance of obesity-induced HCC in males compared to females*'.

Major comments

I find the described mechanism a very complicated one: it details a complex, intertwined signal regulation involving IL6-induced expression of CCRK through phosphorylation of STAT3 and upregulation of the Androgen Receptor (AR)...Aside from CCRK, the novelty of these findings is mild to moderate, as the story proposes a hypothesis that could piece the puzzle together, using as a basis, what is already known in the literature to be involved in NASH and HCC.

Response:

To facilitate the comprehension of the mechanism, we have added a schematic diagram to depict the inflammatory-CCRK circuitry (new **Figure 3J**). Up to now, it is unclear how the hepatic IL-6-STAT3 cascade is activated and sustained during malignant transformation in NASH-associated HCC. Besides, the upstream kinase that controls GSK3 β /mTORC1 signaling in the obesity-induced inflammatory microenvironment has not been elucidated. Our findings suggest that CCRK functions as a major signaling hub in obesity-associated hepatocarcinogenesis, providing insights into therapeutic strategies to reduce tumor burden from the worldwide obesity epidemic.

Minor comments:

1. *The "n" number of samples should be listed per group in each experiment.*

Response:

The number of samples per group have been listed in each experiment in the revised version.

2. *Student's t-test was employed to compare 2 groups even when more than 2 groups were included in the experimental results; this is not statistically appropriate. Even when "comparing" 2 groups, if more than 2 are within the same experimental set-up (or graphs),*

ANOVA should be performed, followed by a post-hoc test, such as Newman-Keuls or Bonferroni, which would then compare any 2 groups of interest. This is applicable to all figures. For example: 1C-F, 1H-J, 2C, 2E-K etc...

Response:

We have repeated the statistical analyses using ANOVA followed by Bonferroni post-hoc test. Revisions were made in Fig. 1C-F, 1H-J, 2C, 2E-K, 5B, 5D, 5I-J, 6D-E, 6G, 6I-J.

3. When assessing glucose tolerance or insulin tolerance by GTT or ITT, quantitative graphs should represent the "area under the curve", Not a picked time-point of interest. The authors, for instance, chose 15min timepoint (Fig. 1D, E and 2K). It is obvious that there are very mild (if any) changes in glucose tolerance or insulin tolerance in Figures 1E and 2K.

Response:

We have re-analyzed the IPGTT and IPITT data using area under the curve (AUC) as suggested. New Figure 1D-E and 2K are shown in the revised version, depicting significant changes in glucose tolerance or insulin tolerance.

4. A bit more detail is required to describe the Figure legends, in general especially when not clarified in the text: for example, describe the samples in Fig. 3L, 4C, 4D and 7B.

Response:

We have revised all figure legends with more details, including those for Figure 3I, 4C-D, 7B as specified by the reviewer.

We hope that this extensively revised manuscript can now be considered acceptable for publication. Thank you very much.

References:

1. Rebouissou S, Zucman-Rossi J, Moreau R, Qiu Z, Hui L. Note of caution: Contaminations of hepatocellular cell lines. *Journal of hepatology* **67**, 896-897 (2017).
2. Tai Y, *et al.* SK-Hep1: not hepatocellular carcinoma cells but a cell model for liver sinusoidal endothelial cells. *Int J Clin Exp Pathol* **11**, 2931-2938 (2018).
3. Poisson J, *et al.* Liver sinusoidal endothelial cells: Physiology and role in liver diseases. *Journal of hepatology* **66**, 212-227 (2017).
4. Zhu J, *et al.* Resistance to cancer immunotherapy mediated by apoptosis of tumor-infiltrating lymphocytes. *Nature Commun* **8**, 1404 (2017).
5. Condamine T, *et al.* Lectin-type oxidized LDL receptor-1 distinguishes population of human polymorphonuclear myeloid-derived suppressor cells in cancer patients. *Sci Immunol* **1**, (2016).

6. Brunt EM, *et al.* Nonalcoholic fatty liver disease (NAFLD) activity score and the histopathologic diagnosis in NAFLD: distinct clinicopathologic meaning. *Hepatology* **53**, 810-20 (2011).
7. Feng H, *et al.* A CCRK-EZH2 epigenetic circuitry drives hepatocarcinogenesis and associates with tumor recurrence and poor survival of patients. *Journal of hepatology* **62**, 1100-1111 (2015).
8. Darlington GJ, *et al.* Expression of liver phenotypes in cultured mouse hepatoma cells. *J Natl Cancer Inst* **64**, 809-19 (1980).
9. Zhou J, *et al.* Hepatoma-intrinsic CCRK inhibition diminishes myeloid-derived suppressor cell immunosuppression and enhances immune-checkpoint blockade efficacy. *Gut* **67**, 931-44 (2018).
10. Yu Z, *et al.* Cell cycle-related kinase mediates viral-host signalling to promote hepatitis B virus-associated hepatocarcinogenesis. *Gut* **63**, 1793-1804 (2014).
11. Stauffer JK, *et al.* Coactivation of AKT and beta-catenin in mice rapidly induces formation of lipogenic liver tumors. *Cancer research* **71**, 2718-2727 (2011).

Reviewers' Comments:

Reviewer #1:

Remarks to the Author:

The authors have addressed the concerns raised by the reviewers. The paper is now significantly improved.

Reviewer #2:

Remarks to the Author:

The authors addressed most of the questions and concerns I had raised. Significant improvements have been made in the revised manuscript to justify the publication.

Reviewer #3:

Remarks to the Author:

The authors have mostly addressed previous concerns and the manuscript is improved. I have a few comments.

1. Data added in Supplementary figure 2E-H are interesting and important. Please add the author's response to my comment 5, "Collectively, these data suggest that CCRK-induced liver steatosis in obesity-associated hepatocarcinogenesis is a net result of increased lipid synthesis and uptake, and decreased lipid secretion.", in the main text.

On the other hand, the authors showed that liver-specific knockdown of CCRK decreased the expression of CPT2 in Supplementary figure 2H, which possibly inhibited fatty acid beta-oxidation. Recently, downregulation of CPT2 has been reported to be rather beneficial for HCC development (Fujiwara N, Gut. 2018 Aug;67(8):1493-1504.; Lin M, Onco Targets Ther. 2018 May 25;11:3101-3110.). The authors should discuss this point by citing these papers. I think that the beta-oxidation in the hepatocytes was decreased as a result of decreased amount of lipids by knockdown of CCRK.

2. I asked whether CCRK Tg mice develop spontaneous liver steatosis in comment 7. Although the authors showed increased liver steatosis in CCRK Tg mice under HFHC diet feeding in Supplementary Figure 4, the histology of the liver under normal chow was not shown. Please add this data.

Reviewer #4:

Remarks to the Author:

The authors have addressed my concerns and the revised manuscript has been significantly improved.

We would like to thank you for providing us with the opportunity to finalize our manuscript. Please see our response to Reviewer#3 as follows:

Comments:

1. Data added in Supplementary figure 2E-H are interesting and important. Please add the author's response to my comment 5, "Collectively, these data suggest that CCRK-induced liver steatosis in obesity-associated hepatocarcinogenesis is a net result of increased lipid synthesis and uptake, and decreased lipid secretion.", in the main text.

On the other hand, the authors showed that liver-specific knockdown of CCRK decreased the expression of CPT2 in Supplementary figure 2H, which possibly inhibited fatty acid beta-oxidation. Recently, downregulation of CPT2 has been reported to be rather beneficial for HCC development (Fujiwara N, Gut. 2018 Aug;67(8):1493-1504.; Lin M, Onco Targets Ther. 2018 May 25;11:3101-3110.). The authors should discuss this point by citing these papers. I think that the beta-oxidation in the hepatocytes was decreased as a result of decreased amount of lipids by knockdown of CCRK.

Response:

We have incorporated these statements and new references in the main text (page 10).

2. I asked whether CCRK Tg mice develop spontaneous liver steatosis in comment 7. Although the authors showed increased liver steatosis in CCRK Tg mice under HFHC diet feeding in Supplementary Figure 4, the histology of the liver under normal chow was not shown. Please add this data.

Response:

We have included the histology of livers from CD-fed control and CCRK TG mice in the new Supplementary Figure 4A.

Yours sincerely,